# Cell surface localisation of GPI-anchored receptors in *Trypanosoma brucei*

Sourav Banerjee[1†‡], Nicola Minshall[1†], Alexander D Cook[2,3], Olivia Macleod[1], Helena Webb[1], Matthew K Higgins[2,3], Mark Carrington[1*]

[1]Department of Biochemistry, University of Cambridge, Cambridge, United Kingdom; [2]Department of Biochemistry, University of Oxford, Oxford, United Kingdom; [3]Kavli Institute for Nanoscience Discovery, Dorothy Crowfoot Hodgkin Building, University of Oxford, Oxford, United Kingdom

*For correspondence:
mc115@cam.ac.uk

†These authors contributed equally to this work

Present address: ‡Division of Medical Research, SRM Medical College Hospital and Research Centre, Faculty of Medicine and Health Sciences, SRM Institute of Science and Technology, Kattankulathur, Tamil Nadu, India

Competing interest: The authors declare that no competing interests exist.

## eLife Assessment

This **valuable** manuscript investigates the localisation of nutrient receptors in bloodstream stage trypanosomes, with implications for both nutrient uptake and immune evasion. Results after direct fixation of the cells in culture medium (as opposed to fixation after centrifugation) provide **compelling** evidence that the amounts of receptors on the surface of the cell, as opposed to the flagellar pocket, have previously been severely underestimated.

**Abstract** *Trypanosoma brucei*, the causal agent of Human and Animal African trypanosomiasis proliferates in the extracellular milieu of mammals. It acquires host macromolecular nutrients by receptor-mediated endocytosis. The best characterised cell surface receptor is for transferrin (TfR), and it has been reported to be preferentially localised in the flagellar pocket domain of the plasma membrane, the sole site of endocytosis. In this location, the TfR may be inaccessible to adaptive immune system effectors. The *T. brucei* genome encodes ~15 TfR variants, and here we compared two, the first attached to the plasma membrane by a single glycosylphosphatidylinositol (GPI)-anchor and the other by two. Transferrin uptake kinetics were similar and rapid for both. Unexpectedly, initial binding of transferrin occurred over the whole cell surface suggesting the TfR was not localised solely in the flagellar pocket. This localisation was confirmed by immunofluorescence assays and was independent of the number of GPI-anchors. Two other GPI-anchored receptors were investigated to determine whether localisation to the whole cell surface was a general property of GPI-anchored receptors. Haptoglobin-haemoglobin uptake assays and immunofluorescence localisation of complement factor H receptor showed both were also whole cell surface localised. The mechanisms by which trypanosome receptors are protected from antibody-mediated attack are more complex than hiding in a pocket.

## Introduction

*Trypanosoma brucei* proliferates within the bloodstream and tissue spaces of mammals and can maintain a long-term infection lasting in some cases for years. To do this, it must avoid the host adaptive and innate immune response at the same time as acquiring nutrients from the host. The adaptive immune response is overcome through a population survival strategy based on a densely packed surface coat of variant surface glycoprotein (VSG) that undergoes antigenic variation. As the antibody titre against one VSG increases, a small subset of trypanosomes that have switched expression to a different VSG escape until they are recognised in turn and so on in an iterative process (*Mugnier et al., 2015*). The trypanosome has evolved mechanisms to increase the titre of host immunoglobulin

required for killing. First, there is rapid endocytosis and degradation of VSG-bound antibody (*Engstler et al., 2007*) and second, a receptor for IgG Fc region (ISG75; *Mikkelsen et al., 2024*). How the innate immune system is countered is less well understood (*Pinger et al., 2018*) but receptors for Complement C3, known as ISG65, and Factor H (FHR) have been identified and characterised (*Cook et al., 2024*; *Macleod et al., 2020*; *Macleod et al., 2022*). Acquisition of small molecule nutrients, such as glucose and amino acids, use typical eukaryotic membrane transporters.

Obtaining host macromolecule nutrients poses a more complex problem. First, *T. brucei* has a very wide host range and presumably receptors recognise ligands from a range of host species, this has been shown to be the case for the transferrin (TfR; *Trevor et al., 2019*). Second, as the TfR and haptoglobin haemoglobin receptor (HpHbR; *Lane-Serff et al., 2016*) bind ligands that have similar dimensions to immunoglobulin molecules then why are host antibodies against receptors not sufficient to clear an infection? Some models to account for this suggested that the VSG coat acts as a barrier preventing immunoglobulin access. However, it was shown that antibodies could penetrate a significant distance into the VSG coat (*Hsia et al., 1996*) but not to the small plasma membrane proximal VSG domain (*Schwede et al., 2011*). Both the ligand size and the penetration of antibodies into the VSG coat made it unlikely that receptors were sterically protected from antibody binding. Immunisation of mice with recombinant ISG65 or ISG75 did not protect against infection (*Ziegelbauer and Overath, 1993*). In addition, accessibility of the HpHbR was demonstrated by the ability of a monoclonal antibody drug conjugate to clear infections in a mouse model (*MacGregor et al., 2019*).

The trypanosome flagellar pocket is an invagination of the plasma membrane at the base of the single flagellum and is the sole site of endo- and exocytosis. The location of the flagellar pocket at the posterior end of the cell is central to the mechanism of clearance of immunoglobulin molecules bound to VSG. The forward swimming motion of the trypanosome exerts a hydrodynamic force on the immunoglobulin that protrudes above the top of the VSG coat pushing the antibody-VSG complex towards the flagellar pocket and subsequent endocytosis (*Engstler et al., 2007*). Since the hydrodynamic flow requires only binding to a cell surface protein, it is likely to occur with any large ligand bound to a receptor providing it has free lateral movement in the plane of the membrane. The flagellar pocket is a discrete domain of the plasma membrane (*Gadelha et al., 2009*) with several exclusively localised proteins (*Gadelha et al., 2015*). At the neck of the pocket, there is a restriction introduced by close contact between the plasma membranes of the flagellum and pocket as it emerges onto the cell body. This flagellar pocket collar is maintained by specific proteins (*Bonhivers et al., 2008*; *Gadelha et al., 2009*). The exclusive localisation of some proteins to the plasma membrane of the flagellar pocket, others to the cell body, and a range to both indicates that there is regulated passage between the two domains of the plasma membrane.

There is variation in the reported localisation of receptors to different domains of the plasma membrane. TfR was the first receptor to be identified, it is glycosylphosphatidylinositol (GPI)-anchored and has been localised to the flagellar pocket and endosomal membrane in some studies but others suggested it was on the flagellar pocket, cell body plasma membrane, and endosomal membrane (*Duncan et al., 2024*; *Mussmann et al., 2003*; *Mussmann et al., 2004*; *Steverding et al., 1994*; *Steverding et al., 1995*; *Tiengwe et al., 2017*). The Complement Factor H receptor (FHR) is a monomer probably GPI-anchored and is localised to the cell body and flagellar plasma membrane (*Macleod et al., 2020*). HpHbR, also probably GPI-anchored, has been localised to the flagellar pocket (*Vanhollebeke et al., 2008*). Reporter proteins with a single GPI-anchor are also present on the cell body plasma membrane (*Schwartz et al., 2005*). ISG65 and ISG75 are type I transmembrane proteins and localise to the flagellar, cell body, and pocket plasma membrane as well as the endosome (*Leung et al., 2011*; *Ziegelbauer et al., 1992*). The localisation of receptors to the flagellar pocket has been proposed to protect the cell from downstream effects of antibody recognition of the receptor but the discovery of new receptors, such as Complement C3 binding by ISG65 (*Macleod et al., 2022*) and FHR (*Macleod et al., 2020*) that are more abundant and distributed over the cell surface seems to contradict this argument.

Newly synthesised or recycled proteins are added to the plasma membrane at the flagellar pocket. Some proteins are retained in the flagellar pocket and have limited or no distribution over the cell surface (*Billington et al., 2023*). It is assumed that these proteins have a retention and/or retrieval signal. For transmembrane proteins a signal could be in the cytoplasm. In contrast, GPI-anchored proteins have no cytoplasmic component and recognition must occur on the external face of the

plasma membrane. Any signal for retention has to operate in a background of sequence variation in VSGs which makes short amino acid motifs unlikely. TfR is a heterodimer with the two subunits, ESAG6 and ESAG7, forming a single binding site for monomeric transferrin. ESAG6, but not ESAG7, has a GPI-anchor so the TfR is tethered to the plasma membrane by a single anchor in contrast to VSG dimers and trimers which have a GPI-anchor on each subunit. There is experimental evidence that the signal for TfR concentration in the flagellar pocket is the number of GPI-anchors per molecule, a single anchor leading to a greater retention, and more than one anchor ready to escape to the surface of the cell body (*Schwartz et al., 2005*; *Tiengwe et al., 2017*). These experiments manipulated the number of GPI-anchors on TfR, heterodimer with a single anchor on one subunit. Adding an anchor to the second subunit caused a change in localisation to the cell surface and endosome as judged by immunofluorescence. However, conditions that led to a few fold increase in expression of single GPI-anchored TfR, including transferrin shortage and growth at high cell density, resulted in the detection of TfR on the cell body surface (*Mussmann et al., 2004*). Whether the surface-localised TfR is functional remains unproven (*Mussmann et al., 2004*; *Schwartz et al., 2005*).

TfR is encoded by a dispersed gene family with ~15 TfR variants. The experiments here started with a chance observation that one of these encoded a receptor with two GPI-anchors while the others have a single anchor. This was exploited to investigate how the presence of one or two GPI-anchors affected function. We show that the second GPI-anchor has no effect on the kinetics of uptake or processing of transferrin through the endosome. However, the second anchor resulted in ≥1.5-fold increase in steady state receptor levels. Further, transferrin uptake experiments provided evidence that TfR was present over the entire cell surface and this was supported by immunofluorescence experiments. An analysis of the localisation of two other putatively GPI-anchored proteins showed that all were present over the entire cell surface. These experiments demonstrate that the trypanosome expresses a set of receptors over the entire cell surface inferring that there are effective mechanisms to resist antibody mediated recognition of conserved cell surface proteins.

## Results

### The BES7 VSG expression site encodes a transferrin receptor with two GPI-anchors

In *T. brucei*, the active VSG gene is transcribed from a bloodstream form expression site (BES) that is telomere proximal. BESs contain up to 12 other co-transcribed expression site associated genes (ESAGs; *Hertz-Fowler et al., 2008*; *Pays et al., 2001*). TfRs are heterodimers encoded by ESAG6 and ESAG7 (*Chaudhri et al., 1994*; *Salmon et al., 1994*). The ESAG6 and 7 subunits share around 80% amino acid identity although ESAG6 has an additional 24 residues at the C-terminus preceding a GPI-anchor addition signal. Different BESs encode variants of ESAG6 and 7 with 90% or more amino acid sequence identity (*Maier and Steverding, 2008*; *Trevor et al., 2019*). In *T. brucei* Lister 427, the isolate used in these experiments, there are 15 BESs (*Barcons-Simon et al., 2023*) and most were sequenced and annotated after transformation associated recombination cloning of telomeres (*Hertz-Fowler et al., 2008*). In the original annotation, BES7 stands out as it has two copies of ESAG6 but apparently lacked ESAG7 and as such would not be able to make a functional TfR. Here, a fresh analysis of the sequence was performed and indicated that the first 'ESAG6' gene annotated in BES7 actually encoded an ESAG7 with a C-terminal extension and a GPI-anchor typical of an ESAG6. In addition, there was a second ESAG6 gene (ESAG6.2) downstream of the annotated copy (ESAG6.1), the two copies encode proteins with a single difference in 401 residues (M110->V). Phylogenetic analysis of ESAG6 and ESAG7 confirmed that a copy of ESAG7 is indeed present in BES7 along with a single copy of ESAG6 (*Figure 1A*).

The origin of the GPI-anchor attachment signal in BES7 ESAG7 was investigated by comparing the ESAG7 mRNA sequences from 13 different BESs (*Figure 1—figure supplement 1*: relationship between ESAG7 variants). This showed that all the ESAG7 genes contained residual coding sequences for GPI-anchor addition but a set of small deletions between nucleotides 985 and 1068 in the ORF resulted in a frameshift leading to a premature stop codon at the C-terminal end effectively removing the GPI-anchor attachment sequence (*Figure 1—figure supplement 2*: comparison of DNA and translation of ESAG7 from BES1 and BES7). A comparison of BES7 ESAG6 and 7 showed that the C-terminal 120 amino acid sequences are identical and there was a clear prediction for a GPI-anchor

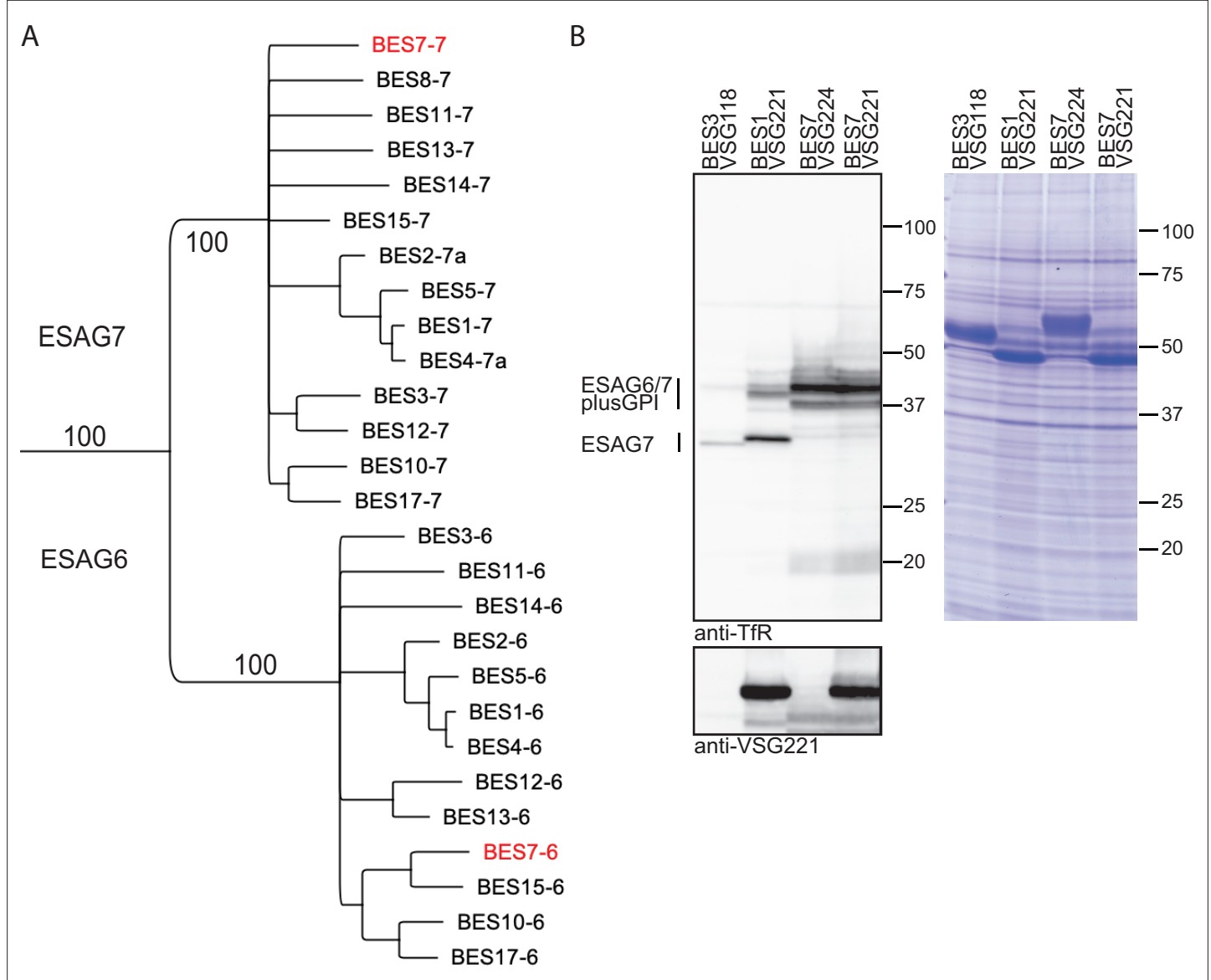

**Figure 1.** *T. brucei* transferrin receptor expressed from BES7 contains GPI-anchored ESAG7. (**A**) Phylogenetic tree showing the relationship between ESAG6 and ESAG7 proteins from *T. brucei* Lister 427 (**Hertz-Fowler et al., 2008**). The tree was assembled in Geneious Prime (https://www.geneious. com) using an alignment by Muscle 5.1 followed by a neighbour joining tree with 100 bootstraps and no outgroup on residues 1–332 of the alignment. BES7 ESAG 6.1 and 7 are shown in red. (**B**) Variation in transferrin receptor expression in cell lines expressing VSGs from different bloodstream form expression sites (BES). BES3-VSG118, BES1-VSG221, BES7-VSG224 cell lines are unmanipulated and the BES7-VSG221 was made after exchanging VSG224 for VSG221 (see *Figure 2A*). Cell lysates were treated with PNGaseF and 2x10⁶ cells equivalents were loaded. Western blots were probed with anti-TfR, and then overprobed with anti-VSG221. ESAG6 and ESAG7 plus GPI-anchor variants are indicated; BES3-VSG118 and BES1-VSG221 cell lysates contain ESAG7 without the GPI anchor. BES3-VSG118 and BES7-VSG224 cell lysates did not show any VSG221. The presence of VSG221 protein confirmed the switching of BES7 VSG224 to VSG221. Switching in VSG type did not affect the expression pattern of native TfR (ESAG6/7 plus GPI) protein. Coomassie-Blue-stained SDS-PAGE gel to demonstrate loading of samples.

The online version of this article includes the following source data and figure supplement(s) for figure 1:

**Source data 1.** Original blots/gels for *Figure 1*.

**Source data 2.** Labelled original Coomassie-Blue-stained gel and membranes corresponding to *Figure 1B*.

**Figure supplement 1.** Relationship between ESAG7 variants, DNA sequence of ESAG7 ORFs from ~base 840.

**Figure supplement 2.** Relationship between ESAG7 variants, comparison of DNA and translation of ESAG7 from BES1 and BES7.

**Figure supplement 3.** Relationship between ESAG 6 and 7 from BES7.

on both using NetGPI 1.1 (*Gíslason et al., 2021*; *Figure 1—figure supplement 3*: alignment of ESAG 6 and 7 from BES7).

## Cellular TfR protein levels are determined by BES identity and the number of GPI-anchors

ESAG6 and 7 protein levels were examined by western blotting in cell lines expressing VSGs and TfRs from three different BESs: BES3-VSG118, BES1-VSG221 and BES7-VSG224 (*Figure 1B*; *Figure 1—source data 1 and 2*). TfRs have several N-linked glycosylation sites and whole cell lysates were treated with PNGaseF to simplify analysis. First, in BES7 cells there was a shift in the apparent molecular weight of ESAG7 from ~30 kDa to ~38 kDa, which is consistent with the additional 24 residues present in ESAG6 but not ESAG7 and a GPI-anchor (*Figure 1B*), such a change in mobility on the addition of a GPI-anchor to ESAG7 has been reported previously (*Tiengwe et al., 2017*). Second, the identity of the active BES affected both ESAG6 and ESAG7 protein levels, there were increasing levels of expression: BES3 <BES1<BES7 (*Figure 1B*). Lastly, when the BES7 cell line was modified to exchange VSG224 to VSG221 (see below), this had little effect on TfR expression, suggesting the BES identity was the dominant determinant rather than the VSG type. The rabbit TfR antiserum used was raised against the TfR from BES1 (*Gerrits et al., 2002*) and is the same as used in most other studies (*Mussmann et al., 2003*; *Tiengwe et al., 2017*). The amino acid sequence identity between BES2 TfR and BES1 TfR is >98% and BES7 TfR is >92% and it is unlikely that the difference in TfR levels result from differential reactivity of the serum to different TfRs. Support for the different levels of TfR expression was obtained when the levels of TfR mRNAs were estimated in the BES3, BES1, and BES7 cell lines using RNAseq (*Supplementary file 1*). The level of TfR mRNAs were expressed as a ratio (ESAG6 +ESAG7):VSG and were consistent with the differences seen in protein levels (BES3, 0.022<BES1, 0.038<BES7, 0.077).

We next investigated whether GPI-anchor number contributed to the variation in cellular TfR levels by making a set of transgenic cell lines (*Figure 2A*). First, VSG224 in BES7 was changed to VSG221. Next, the cell line expressing VSG221 from BES7 was modified to alter the ESAG7 by exchanging the 3' end of the ORF with that from BES1 ESAG7 resulting in the loss of the GPI-anchor signal. As a control, a second cell line in which the ESAG7 gene retained the GPI-anchor addition sequence was made. Similar modifications were performed on cells expressing VSG221 from BES1, cell lines expressing either BES1 ESAG7 modified with the C-terminal GPI-anchor sequence from BES7 ESAG7 or the endogenous BES1 ESAG7 were made. The result was four cell lines: all expressed VSG221 from either BES1 or BES7 and TfR with or without a GPI-anchor on ESAG7.

The cell lines were characterised by western blotting of PNGaseF-treated whole cell lysates (*Figure 2B*; *Figure 2—source data 1 and 2*). The addition of a GPI-anchor signal sequence to ESAG7 resulted in an increase in both ESAG6 and 7. To estimate this change, whole cell lysates were minimally separated by SDS-PAGE to collapse the ESAG6 and 7 proteins into a single band prior to western blotting. Relative quantitation was obtained using a titration curve of the highest TfR expresser and normalised against the cytoplasmic protein SCD6 for loading (*Figure 2C*). Three biological replicates were performed and the levels of both TfRs from BES1 are significantly lower (p<0.01), ~3.5-fold, compared to BES7, regardless of GPI-anchor number. The presence of a GPI-anchor on ESAG7, leading to a double-anchored TfR, significantly increased (p<0.05) the expression level by approximately 1.9-fold in BES1 and about 1.5-fold in BES7. This provides strong evidence that the presence of a GPI-anchor on ESAG7 contributes to the higher cellular expression level of TfR, which could result from increased protein stability, reported for VSGs (*Bangs et al., 1986*; *Engstler et al., 2004*), or a higher rate of turnover and/or greater shedding of the single GPI-anchored TfR (*Schwartz et al., 2005*).

## Transferrin uptake kinetics are similar in cells expressing TfRs with either one or two GPI-anchors

To investigate whether anchor number influenced function, BES7-VSG221 cell lines expressing single or double GPI-anchored TfR were used. It has already been shown that a TfR with double GPI-anchors is functional (*Tiengwe et al., 2017*), here the kinetics of transferrin binding and uptake by single and double GPI-anchor TfRs were compared. The experimental approach was similar to that developed to characterise the binding and endocytosis of VSG antibodies (*Engstler et al., 2007*). The medium

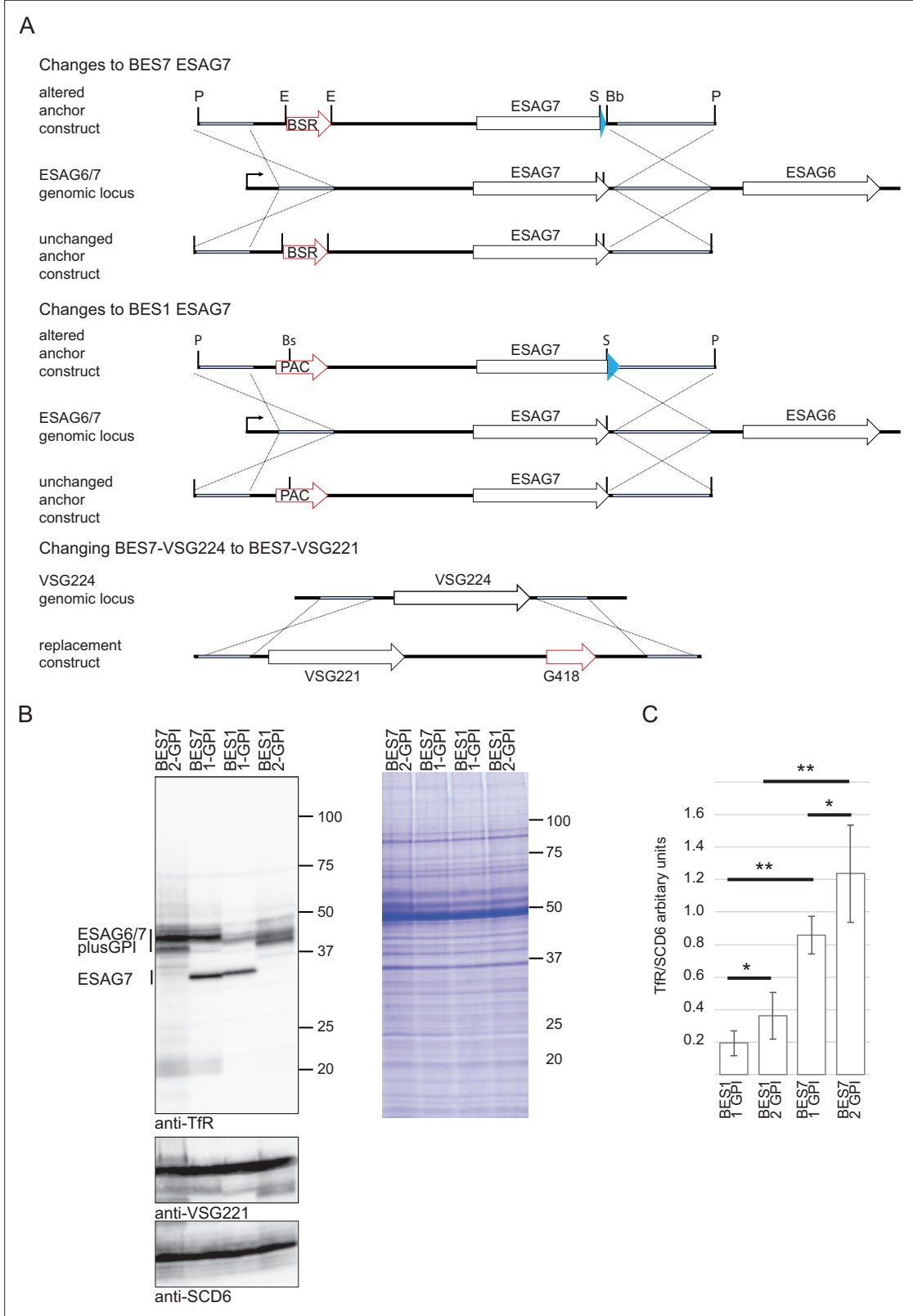

**Figure 2.** The effect of BES identity and the number of GPI-anchors on the cellular TfR levels. (**A**) Schematic illustrating the production of cell lines. For the addition and removal of ESAG7 GPI-anchor addition sequences, two constructs were used for each of BES1 and BES7. A description of the construction is provided in the methods. For BES7, the same process was used, one construct removing the GPI-anchor from ESAG7, the other leaving it intact. For BES1, integration of one resulted in modification of ESAG7 to add a GPI-anchor, integration of the second resulted in an unchanged ESAG7

*Figure 2 continued on next page*

*Figure 2 continued*

polypeptide sequence. Restriction enzyme sites used in cloning: P, PacI; E, EcoRI; S, SacI; Bs, BstEII; and Bb, BsrBI. The VSG in BES7-VSG224 cells was switched to VSG221 to remove any VSG-dependent variation. The targeting construct conferred G418 resistance. (**B**) Transferrin receptor expression in the following cell lines: BES7-VSG221 single (1-GPI) and double (2-GPI) GPI-anchored TfR and BES1-VSG221 single and double GPI-anchored TfR. Cell lysates were treated with PNGaseF and $2\times10^6$ cells equivalents were loaded per gel track. A western blot of the samples was probed with anti-TfR then over probed with anti-VSG221 and anti-SCD6. ESAG6 and ESAG7 plus GPI-anchor variants are indicated. (**C**) TfR quantitation estimates from a minimum of three biological replicates of each cell line, error bars display the standard deviation for each sample. Samples plus a dilution series from BES7-221 were minimally separated by SDS-PAGE to compress TfR components to a single band for analysis after western blotting, probed with anti-TfR and anti-SCD6. Donkey anti-rabbit Alexa 488 was used as secondary antibody and band intensity was calculated by ImageJ ensuring that signals taken for each reading were below saturation levels. Relative quantitation was determined by mapping values to those obtained from a dilution series. Measurements for SDC6 were used to normalise cell loading with the value for BES1-1GPI anchor samples given an arbitrary value of 1. Statistical analysis was used a two-tailed paired t-test, $**p\leq0.01$; $*p\leq0.05$.

The online version of this article includes the following source data for figure 2:

**Source data 1.** Original blots/gels for *Figure 2B*.

**Source data 2.** Labelled original Coomassie-Blue-stained gel and membranes corresponding to *Figure 2B*.

used to culture trypanosomes contained ~2.5 µM transferrin (*Kakuta et al., 1997*), Alexa 568 labelled bovine holo-transferrin (Tf$_{A568}$) was added to cultures, without any other manipulation, to a final concentration of 0.5 µM (so that one in six molecules of transferrin in the medium is now fluorescent) and cells were then fixed over a time course from 15 to 600 s and imaged by epifluorescence microscopy (*Figure 3A*, *Figure 3—figure supplement 1A*: larger fields of view of cells from *Figure 3A*). The on rate for binding of transferrin to TfR is very rapid ($4.5 \times 10^5$ M$^{-1}$ s$^{-1}$ at pH 7.4 for BES1 TfR and bovine transferrin; *Trevor et al., 2019*) and binding will occur to unoccupied receptor almost immediately. At 15 s, there was variation between individual cells with Tf$_{A568}$ present on the flagellum, across the cell body and the flagellar pocket in both cell lines. There was also variation at 60 s, Tf$_{A568}$ was present in the endosomal compartment of most cells, and the cell surface in some cells. At 600 s, the Tf$_{A568}$ was predominantly within the endosomal compartment. The variation was also examined by quantifying the fluorescence from individual cells (*Figure 3B*), this showed the range of variation and an increase in fluorescence between 15 and 60 s followed by a decrease by 600 s. Fluorescence intensity was also determined across transverse sections of the cell (*Figure 3C*), this showed some cells with an internal signal and others with two peaks consistent with a cell surface localisation. There was a stronger signal from cells expressing TfR with two GPI-anchors, consistent with higher expression levels, but no consistent differences in distribution were seen.

The same Tf$_{A568}$ uptake assay was performed with the BES1-VSG221 cells expressing TfRs with one or two GPI-anchors (*Figure 3D*, *Figure 3—figure supplement 1B*: larger fields of view of cells from *Figure 3D*). There was a reduced binding and uptake of Tf$_{A568}$ in BES1 cells compared to BES7 cells as might be expected given lower levels of the receptor. Otherwise, the time course of uptake was similar: at 15 s Tf$_{A568}$ was present on the flagellum and cell body surface as well as the flagellar pocket in both cell lines, at 60 s Tf$_{A568}$ on the cell surface was reduced and some was now present in the flagellar pocket and endosomal compartments, and by 600 s Tf$_{A568}$ was substantially within endosomal compartments. In both BES7 and BES1 cell lines, fluorescence distribution patterns across the cells showed variability at 15 and 60 s (*Figure 3E–F*), but appeared more uniform at 600 s.

Previous studies have shown that some TfRs do not bind canine transferrin (*Mussmann et al., 2003*). Canine transferrin has 501/686 amino acid sequence identities with bovine transferrin (*Figure 3—figure supplement 2A*: Alignment of bovine and canine transferrin) and it was used as a negative control in the same transferrin binding and uptake assay above using BES7-VSG221 cells expressing TfR with either one or two GPI-anchors (*Figure 3—figure supplement 2B*: Canine transferrin control for *Figure 3*). No cell binding or uptake of canine transferrin in either cell line was observed. In addition, a competition assay for binding of fluorescent Tf$_{A568}$ was performed by adding 5- or 10-fold excess of unlabelled transferrin at the same time (*Figure 3—figure supplement 3*: Unlabelled competition control for *Figure 3*). The signal from Tf$_{A568}$ was reduced by the competition indicating saturable binding, consistent with a receptor mediated process. These findings provide evidence that the binding of bovine transferrin to the cell surface is receptor dependent. An important point is that fluorescent transferrin was added to cells in culture without any other manipulation as preliminary experiments indicated that centrifugation, washing, and

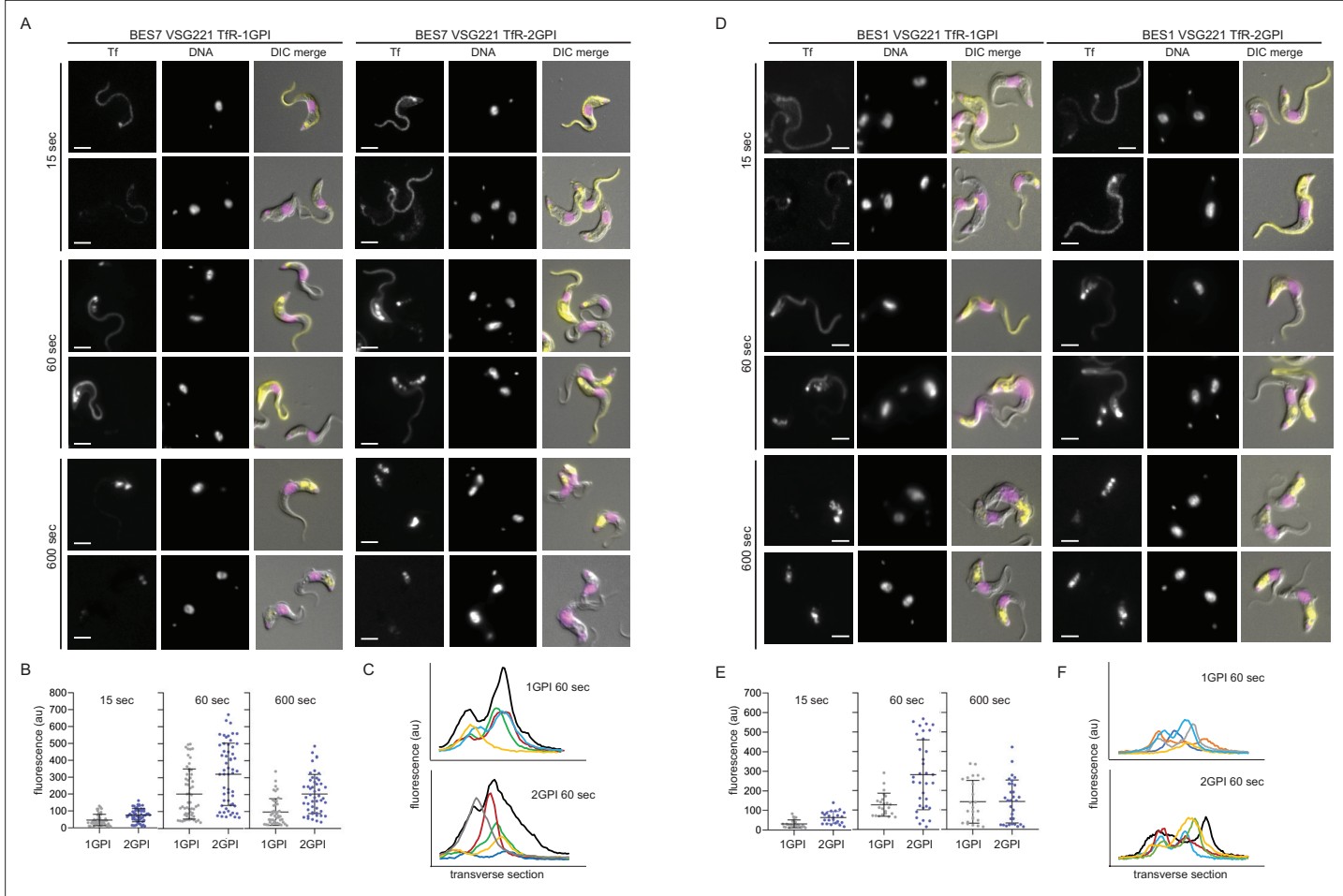

**Figure 3.** Transferrin endocytosis is rapid, in both single and double GPI-anchored cells. Initial binding is to the cell surface and internalisation is detected within 60 s. Cultures of single (1-GPI) or double (2-GPI) GPI-anchored TfR expressing BES7-VSG221 and BES1-VSG221 cells were supplemented with 500 nM Alexa568-labelled holo-transferrin (Tf$_{A568}$) and incubated for 15, 60, or 600 s and fixed by addition of formaldehyde for 5 min. Following washing with PBS containing 1% bovine serum albumin cells were visualised under Zeiss Axio Imager.Z2 with ×100 objective. Images in all these experimental time points for both 1- and 2GPI TfR cells were captured with same intensity of excitation and exposure time. (**A**) Images showing the location of bound or endocytosed transferrin in representative cells in populations and at time points as indicated. Scale bar is 5 µm. Images were processed with ZEN Blue 3.4 software without any deconvolution. Larger numbers of cells are shown in *Figure 3—figure supplement 1*. (**B**) The variation between individual cells in transferrin binding and uptake and endocytosis shown by measurements of total fluorescence intensity of Tf$_{A568}$ in BES7 VSG221 1- and 2-GPI TfR cells at 15, 60, and 600 s. Measurements were obtained using Zen Blue 3.4 software and normalising the background fluorescence (*Supplementary file 3*). Scatter plots of measurements of individual cells are shown for 1-GPI (grey) and 2-GPI (purple) TfRs. The average and standard deviation are shown for each cell line. (**C**) Tf$_{A568}$ fluorescence intensity distribution across transverse sections of randomly selected cells was measured using ImageJ software. The plots are shown for both BES7 1- and 2-GPI cell lines, showing the pattern of individual cells in different colours at 60 and 600 s. (**D**), (**E**) and (**F**) same as (**A**) (**B**) and (**C**) but with BES1-VSG221 1- and 2GPI TfR expressing cells (*Supplementary file 4*).

The online version of this article includes the following source data and figure supplement(s) for figure 3:

**Figure supplement 1.** Transferrin endocytosis is rapid, in both single and double GPI-anchored cells.

**Figure supplement 2.** Control for specificity of bovine transferrin binding to *T. brucei* BES7 VSG221 transferrin receptors.

**Figure supplement 3.** Control for specificity of bovine transferrin binding to *T. brucei* BES7 VSG221 transferrin receptors.

**Figure supplement 4.** Changes in cellular TfR levels caused be washing.

**Figure supplement 4—source data 1.** Original membranes.

**Figure supplement 4—source data 2.** Labelled original membranes.

resuspension in medium led to a reduction in cellular TfR for up to 2 hr (*Figure 3—figure supplement 4*: Changes in cellular TfR levels caused by washing *Figure 3—figure supplement 4—source data 1 and 2*).

Together, these experiments support a model in which the TfR is present on the cell surface, and endocytosis is a rapid process. The greater transferrin binding and uptake in BES7 compared to BES1 cells provides evidence that TfR abundance limits the amount of transferrin endocytosed. The number of GPI-anchors on the TfR has no obvious influence on transferrin uptake but does affect receptor levels.

The finding that $Tf_{A568}$ was bound over the whole cell surface at early time points in the uptake assays suggested a similar localisation for the TfR. This is not consistent with some previous studies that found TfR and other GPI-anchored receptors sequestered in the flagellar pocket (*Duncan et al., 2024*; *Mussmann et al., 2004*; *Steverding et al., 1994*; *Steverding et al., 1995*; *Tiengwe et al., 2017*; *Vanhollebeke et al., 2008*) or that expression level determined localisation (*Mussmann et al., 2003*). We went on to characterise the cellular localisation of single and double GPI-anchored TfR along with two other GPI-anchored receptors.

## Localisation of the transferrin receptor by immunofluorescence

Previous reports provided evidence that in BES1-VSG221 cells grown in the presence of 10% FBS, the cell surface TfR is located or concentrated within the flagellar pocket and endosome, rather than dispersed on the cell body (*Ligtenberg et al., 1994*; *Mussmann et al., 2003*; *Salmon et al., 1994*; *Tiengwe et al., 2017*). In contrast, when cells are treated to increase TfR expression by roughly three to fivefold, for example when grown to high density or in canine serum, TfR localises on the cell body (*Mussmann et al., 2003*). To resolve this discrepancy, the localisation of the TfR was investigated using immunofluorescence. Based on the transferrin uptake experiments, we reasoned that the localisation of the TfR would be very dynamic and thus chose to fix the cells in culture without any centrifugation or washing, this approach was supported by preliminary experiments comparing protocols (*Figure 4—figure supplement 1*: the effect of fixation conditions on TfR localisation by immunofluorescence) that showed both the composition of the fixative and absence of a washing step altered the outcome.

Four cell lines, BES1-VSG221 and BES7-VSG221 cells each expressing TfR with one or two GPI-anchors, were investigated. Fixative was added to cells in culture without any other manipulations, the final concentrations were 4% formaldehyde and 0.2% glutaraldehyde and TfR localisation was observed by immunofluorescence after settling cells on microscope slides. Under these experimental conditions, the TfR in all four cell lines had a distribution across the entire cell surface including the flagellar pocket and in internal compartments in some cells (*Figure 4A*). Fluorescence intensity measurements across transverse sections of both the BES7 1- and 2-GPI cells confirmed that TfR is indeed present on the surface of some cells as seen by two peaks in fluorescence corresponding to the cell periphery in some cells (*Figure 4B*). As in the transferrin uptake experiments, the strength of signal as measured by total fluorescence intensity from individual cells showed that TfR abundance varied between individual cells in populations (*Figure 4C*, *Figure 4—figure supplement 2*: larger fields of view of cells in *Figure 4C*). These measurements also showed that the presence of two GPI-anchors on TfR increased levels by about 1.5-fold compared to the single anchored receptor and that there is an ~3.5-fold higher TfR expression in BES7 compared to BES1 cells regardless of GPI-anchor number. These measurements support the western blot quantitation (*Figure 2*) and also provide evidence that the immunofluorescence signal does not contain a significant level of non-specific background.

Structured Illumination Microscopy (SIM) was used to obtain more information on TfR localisation in the same four cell lines as above (*Figure 5*). TfR was present on the cell surface including the flagellar pocket, cell body and flagellum, and in the endosome (single stack images; *Figure 5A*). Three-dimensional reconstruction of the localisation of TfR confirmed that it is distributed across the cell surface and in endosomes (*Figure 5B*).

These experiments collectively provide strong evidence that TfR is indeed present on the *T. brucei* cell surface under normal growth conditions and that the localisation pattern is independent of the presence or absence of a GPI-anchor in the ESAG7 subunit.

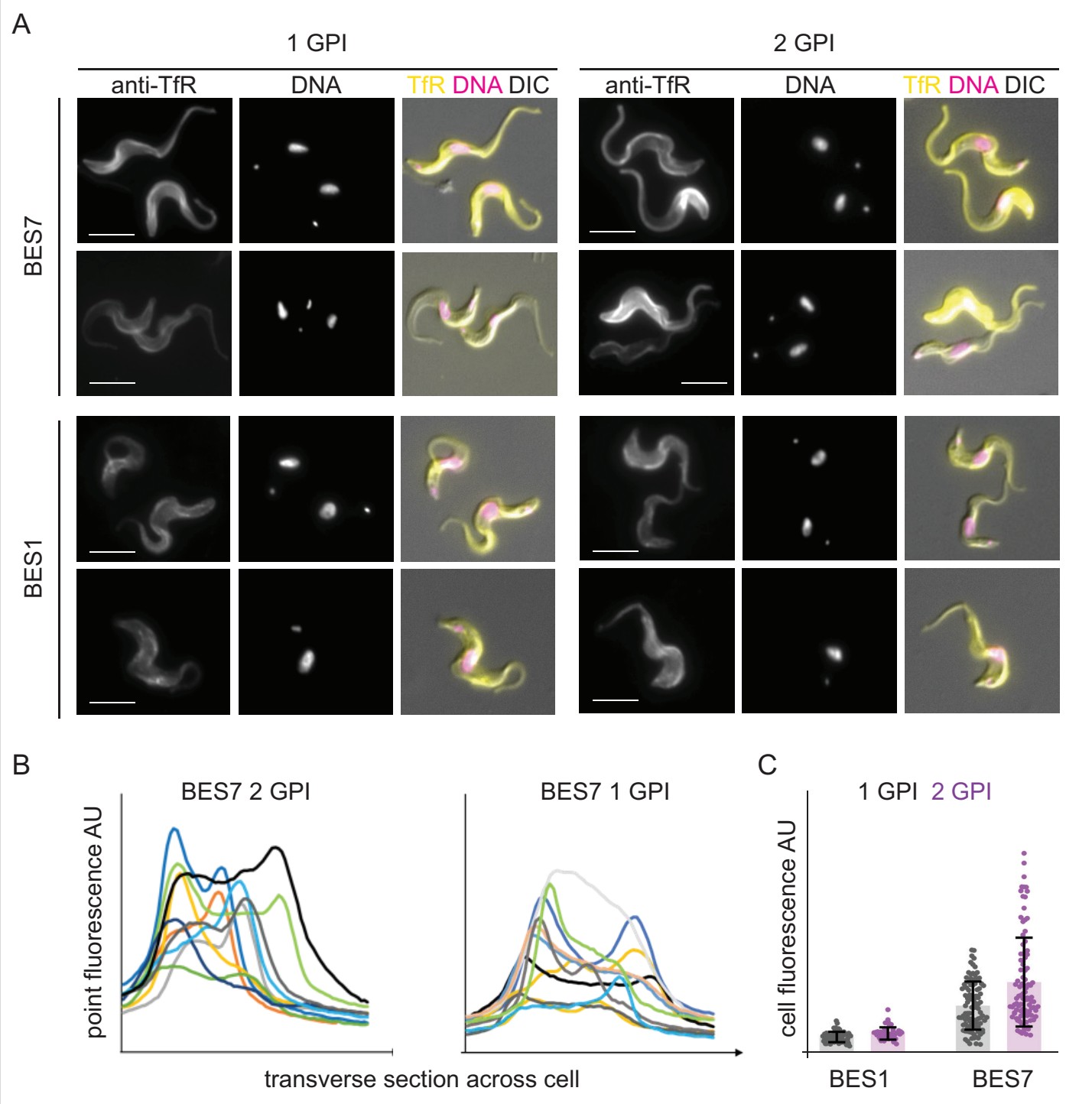

**Figure 4.** Immunofluorescence localisation of single and double GPI-anchored transferrin receptors to the entire plasma membrane. Cell lines expressing VSG221 and either 1- or 2-GPI TfR variants were fixed in culture with final concentrations of 4% formaldehyde and 0.2% glutaraldehyde. Immunofluorescence localisations were performed with rabbit anti-TfR followed by donkey anti-rabbit Alexa 488 (green). Cells were visualised using a Zeiss Axio Imager.Z2 with X100 objective. Images in all these experimental sets were captured with same intensity of excitation and exposure time. (**A**) Localisation in cell lines as indicated. Scale bar is 5 µm. Images were processed with ZEN Blue 3.4 software. A larger number of cells is shown in *Figure 4—figure supplement 2*. (**B**) Fluorescence distribution across transverse sections of the cells was measured using ImageJ software. Plots are shown for both BES7-VSG221 TfR cell lines, each cell in a different colour. (**C**) Mean anti-TfR fluorescence intensity in individual cells of both BES1 VSG221 and BES7 VSG221 1- and 2-GPI TfR cell population was measured using ZEN Blue 3.4 and normalised against the background fluorescence

*Figure 4 continued on next page*

*Figure 4 continued*

(**Supplementary file 5**). Bar diagram shows the fluorescence intensity of either 1-GPI (grey) or 2-GPI (purple) TfR from individual cells. The columns represent the average and the error bars show the standard deviation.

The online version of this article includes the following figure supplement(s) for figure 4:

**Figure supplement 1.** Immunofluorescence localisation of single and double GPI-anchored transferrin receptors to the entire plasma membrane with different fixation protocols.

**Figure supplement 2.** Immunofluorescence localisation of single and double GPI-anchored transferrin receptors to the entire plasma membrane.

## Cell surface localisation is shared by two other GPI-anchored receptors

The finding that TfR is distributed over the whole cell surface led to an investigation of other receptors. Two were selected on the basis of being predicted to have one or more GPI-anchors and a non-expressing negative control being available. The first was the haptoglobin haemoglobin receptor (HpHbR) for which +/+and -/- cell lines were used. The structure of most of HpHbR is known, and it is a monomer in solution but two or more receptors can bind a single ligand (**Higgins et al., 2013**; **Lane-Serff et al., 2014**). HpHbR has a low copy number of 300–400 per cell (**Vanhollebeke et al., 2008**) and, in the absence of a suitable antibody, a ligand uptake assay was used. The second receptor analysed was the complement factor H receptor (FHR), this was used after a fortuitous observation that although it is not expressed when cells are grown in medium containing 10% foetal bovine serum (**Macleod et al., 2020**), it is expressed at high levels when the same cells are grown in medium with 10% rabbit serum replacing the foetal bovine serum. The structure of FHR is known and it is a monomer with a single GPI-anchor (**Macleod et al., 2020**).

Alexa 568-labelled HpHb (HpHb$_{A568}$) was used in uptake assays with HpHbR +/+ and -/- cells as for transferrin uptake (**Figure 6A**). There was binding of ligand to the cell surface by 15 s, uptake into the endosome by 60 s, and a location consistent with the lysosome by 600 s. Binding and uptake were not detected in HpHbR -/- cells (**Figure 6A**). The kinetics of uptake were similar to those for transferrin. Presence of HpHb$_{A568}$ on the cell surface was confirmed by the fluorescence signal measurement across the cells at 15 s, which showed two distinct peaks consistent with peripheral localisation for some cells (**Figure 6B**), the noise in the quantification is due to the low copy number of the receptor. Again, there was variability in binding and uptake of HpHb across the population (**Figure 6—figure supplement 1**: larger fields of view of cells in **Figure 6**).

In trypanosomes cultured in medium containing 10% rabbit serum, expression of the FHR was readily detected whereas in medium containing foetal bovine serum (FBS) it is not (**Figure 7A**). Cells cultured with rabbit serum were used to determine the localisation of FHR by immunofluorescence following the same procedure used for the TfR (**Figure 7B**). The FHR was clearly localised over the entire cell surface. Two controls for specificity of antibody binding were performed: first cells were grown in medium supplemented with foetal bovine serum and second with a no primary antibody control for the cells grown with rabbit serum to control for cell surface immunoglobulin originating from the growth medium. Neither control gave a signal (**Figure 7—figure supplement 1**: larger fields of view of cells in **Figure 7B** and controls for antibody specificity). SIM analysis further provided spatial resolution along with reconfirming the distribution of the FHR across the trypanosome cell surface and flagellum as well as within the flagellar pocket (**Figure 7C**).

These experiments show that the detection of TfR on the cell surface is part of a general pattern where GPI-anchored receptors are distributed across the cell body plasma membrane.

## Discussion

Any pathogen that has evolved to establish and maintain a long-term infection within a mammalian host must be able to acquire nutrients as well as be resistant to both the innate and adaptive immune responses of the host. African trypanosomes have evolved a strategy that is dependent on variant surface glycoprotein (VSG), a superabundant protein that forms a densely packed coat on the cell surface. Low frequency switching to a different VSG allows the population to persist (**Mugnier et al., 2015**). The VSG coat appears to prevent antibodies, and possibly innate immunity factors, gaining immediate access to the plasma membrane (**Schwede et al., 2011**). However, such a coat cannot be totally impermeable as trypanosomes require host macromolecules such as transferrin and

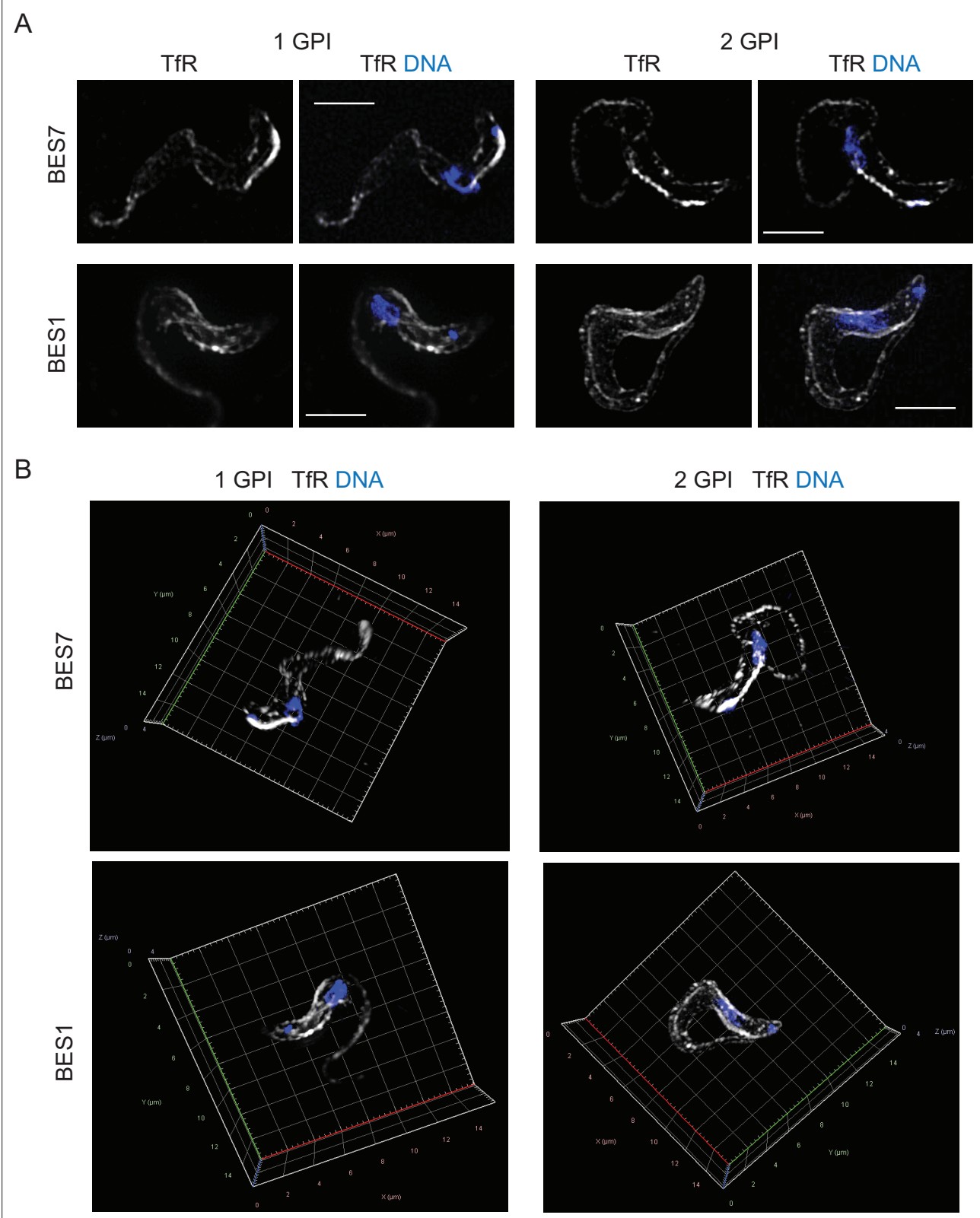

**Figure 5.** Localisation of single and double GPI-anchored transferrin receptors under 3D structured illumination microscope. Cells expressing either 1- or 2-GPI-anchored TfR were fixed in culture with final concentrations of 4% formaldehyde and 0.2% glutaraldehyde. Immunofluorescence localisation was performed with rabbit anti-TfR and donkey anti-rabbit Alexa 488. Cells were visualised using a Zeiss Elyra7 microscope with X63 objective. Images in all these experimental sets were captured with same intensity of excitation and exposure time. (**A**) Localisation of TfR in different cell lines as indicated.

*Figure 5 continued on next page*

Figure 5 continued

Scale bar is 5 µm. (**B**) 3D projection generated using ZEN Blue 3.4 software example cells from the cell lines as indicated. These images show the localisation pattern of TfR on the cell surface as well as within intracellular compartments. X-Y-Z axis is indicated. Scale bar 5 µm.

haptoglobin-haemoglobin which they acquire through receptor mediated endocytosis (*Lane-Serff et al., 2014*; *Salmon et al., 1994*; *Steverding et al., 1994*; *Steverding et al., 1995*; *Trevor et al., 2019*; *Vanhollebeke et al., 2008*).

Why these receptors do not result in the trypanosomes becoming susceptible to immune recognition and attack is central to understanding the mechanisms of infection (*Banerjee et al., 2024*). One model proposed that the receptors are sequestered in the flagellar pocket, an invagination of the plasma membrane at the base of the flagellum, and are not accessible to immune effectors. The idea that sequestration in the flagellar pocket allowed receptors to hide from host immune effectors was first challenged by the discovery of invariant surface glycoproteins (ISGs) (*Ziegelbauer et al., 1992*; *Ziegelbauer and Overath, 1992*) that were shown to be distributed over the whole cell surface (*Macleod et al., 2022*; *Ziegelbauer and Overath, 1993*). Two ISGs were subsequently identified as receptors, one for complement C3 and another for the Fc region of immunoglobulins (*Macleod et al., 2022*; *Mikkelsen et al., 2024*). Both the ability to access ligands and the structures of the extracellular domains of both ISGs indicated that they were unlikely to be physically shielded by the VSG coat. The ISGs are type 1 transmembrane proteins with a large extracellular domain, a single transmembrane pass and a small cytoplasmic tail. In contrast, VSGs have two or more GPI-anchors whereas the two characterised nutrient receptors have a single GPI-anchor. Could this difference in GPI-anchor number result in selective retention of nutrient receptors in the flagellar pocket (*Tiengwe et al., 2017*) as it is difficult to envisage a requirement for any amino acid sequence motifs in the context of variation of VSGs.

Here, we have investigated the distribution of GPI-anchored receptors on the cell surface of mammalian infective *Trypanosoma brucei*. The main findings are: (1) The majority of TfRs encoded in the genome have a single GPI-anchor but there is at least one with two GPI-anchors. (2) The single-anchored TfRs probably arose from loss of one anchor as opposed to gain in the double-anchored version. The loss of the GPI-anchor from ESAG7 may provide a selective advantage possibly reducing exposure to the host adaptive immune system without affecting transferrin acquisition (*Trevor et al., 2019*). (3) The steady-state level of TfR varies with the identity of the transcribed BES. (4) Single- or double-anchored TfRs bind transferrin, as previously reported (*Tiengwe et al., 2017*) and result in similar kinetics of uptake, with movement from the cell surface to the endosome occurring within 2 min. The rapid kinetics of uptake indicates the receptor ligand complex is probably subject to the same hydrodynamic forces as antibody VSG complexes (*Engstler et al., 2007*) resulting in a directional movement towards the posterior end of the cell, the location of the flagellar pocket and endocytosis. (5) Adding a second GPI-anchor to a TfR resulted in a~1.5-fold increase in cellular receptor levels. (6) Both single and double GPI-anchored TfRs are present over the whole cell surface as seen by both transferrin and antibody binding. (7) Two other GPI-anchored receptors, the HpHbR and FHR, both probably with a single GPI-anchor, are present over the whole cell surface.

The earlier findings that the TfR, and subsequently the HpHbR, are localised to the flagellar pocket is an attractive model as it provided an easy explanation for why the receptors did not result in immune attack. The experiments presented here do not agree with this model and it is worth discussing the reasons why these experimental differences have arisen. Here the cells were fixed in culture using formaldehyde and a higher concentration (0.2%) of glutaraldehyde than in previous reports. In contrast, much of the previous work involved manipulations of the trypanosomes before determining either transferrin binding or TfR/ HpHbR localisation by immunofluorescence. The most common is centrifugation and washing prior to fixation for immunofluorescence (*Engstler et al., 2004*; *Roggy and Bangs, 1999*; *Tiengwe et al., 2017*; *Vanhollebeke et al., 2008*), a process that is slow when compared to the very rapid endocytosis of surface-bound transferrin (*Figure 3*) and providing time for the ligand bound receptors to be cleared from the cell surface. This would be even more apparent if the experimental manipulation reduced the export of new or recycled TfRs to the cell surface. An example of how protocols determine outcomes is the variation in the measured half-life of TfR, which ranges from ~40 min to 7 hr depending on the different experimental protocols used (*Biebinger et al., 2003*; *Kabiri and Steverding, 2000*; *Mussmann et al., 2004*; *Tiengwe et al., 2017*).

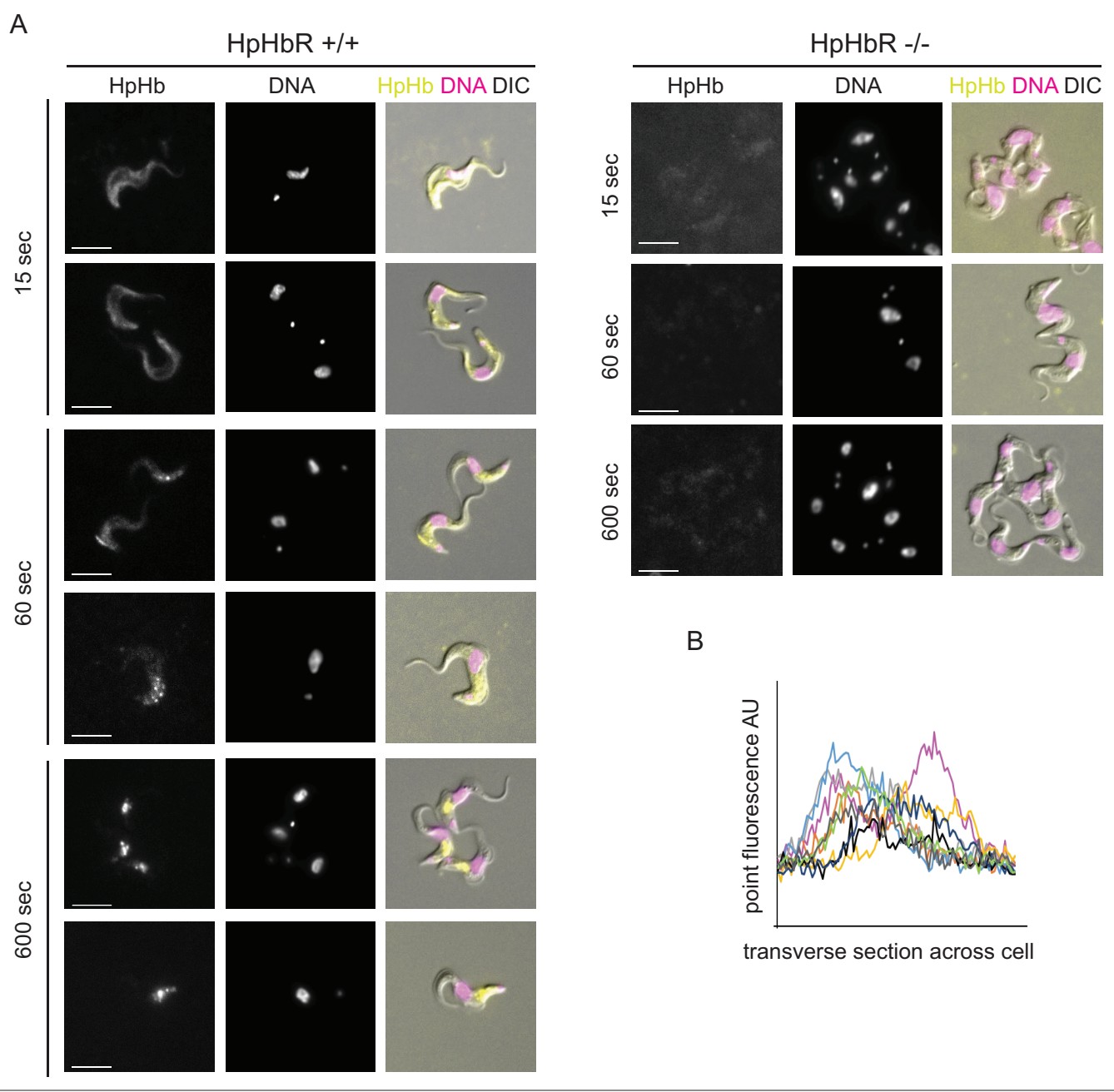

**Figure 6.** Haptoglobin-haemoglobin endocytosis assay. Initial binding is to the cell surface followed by rapid endocytosis. Cells were supplemented with 500 nM Alexa 568 labelled HpHb, incubated for 15, 60, or 600 s and fixed in culture with 1% formaldehyde. Following washing with PBS containing 1% bovine serum albumin. Cells were visualised under Zeiss Axio Imager.Z2 with x100 objective. Images in all these experimental time points were captured with same excitation intensity and exposure time. (**A**) Images showing the location of HpHb in +/+ and -/- cells at different time points in representative cells as indicated. Images were processed with ZEN Blue 3.4 software. Scale bar is 5 µm. A greater number of cells is shown in *Figure 6—figure supplement 1*. (**B**) HpHb fluorescence intensity distribution over transverse section of cells at 15 s measured using ImageJ software. Individual cells are shown in different colours.

The online version of this article includes the following figure supplement(s) for figure 6:

**Figure supplement 1.** Haptoglobin-haemoglobin endocytosis assay.

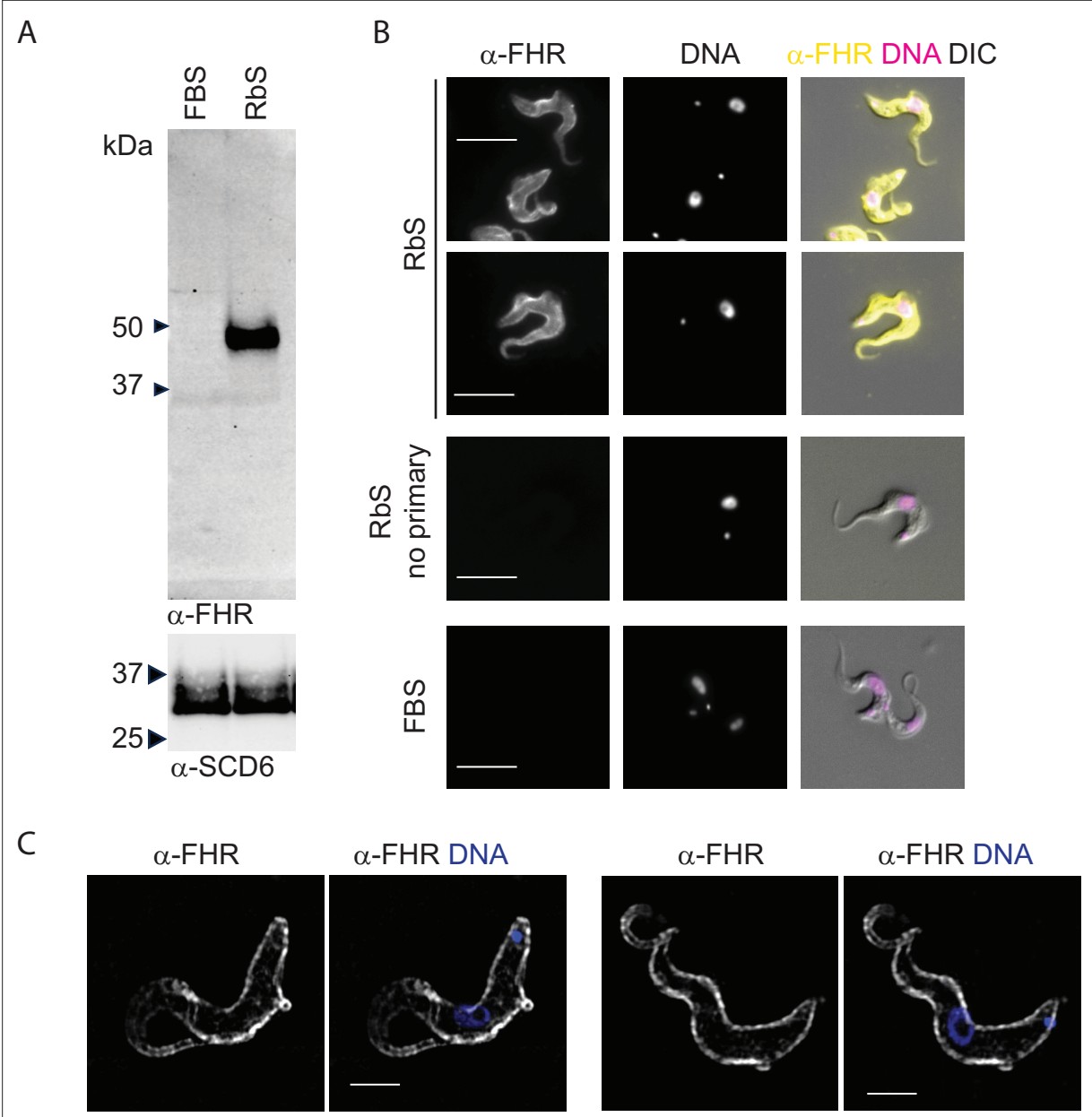

**Figure 7.** The complement factor H receptor localises to the cell surface. The observation that abundant FHR expression occurred in cells grown in rabbit serum but not in FBS supplemented medium was used to investigate its localisation. (**A**) Western blot analysis of FHR expression in *T. brucei* EATRO1125 cells grown in medium supplemented with either foetal bovine serum or rabbit serum. Cell lysates were not treated with PNGaseF and $2\times10^6$ cells equivalent were loaded. The blot was probed with rabbit anti-FHR and then overprobed with anti-SCD6 for loading. (**B**) Immunofluorescence determination of localisation of FHR in cell lines as indicated. Cells were fixed in culture with final concentration of 4% formaldehyde and 0.2% glutaraldehyde. Rabbit anti-FHR antibody was followed by donkey anti-rabbit Alexa 488 secondary antibody. A set of cells grown in RbS, fixed similarly and stained only with secondary antibody was used as a control for residual rabbit IgG from the growth medium and cells grown in FBS for non-expressers. Cells were visualised using a Zeiss Axio Imager.Z2 with x100 objective. Scale bar is 5 µm. A larger number of cells is shown in *Figure 7— figure supplement 1*. (**C**) Localisation of FHR by 3D structured illumination microscopy. Cells fixed as above were visualised using a Zeiss Elyra7 with x63 objective. Images were processed with ZEN Blue 3.4 software. Two representative cell images showing FHR in white and merged panel with DNA in blue. Scale bar is 5 µm.

The online version of this article includes the following figure supplement(s) for figure 7:

**Figure supplement 1.** The complement factor H receptor localises to the cell surface.

The experiments involving transferring BES1-VSG221 cells to medium in which FBS had been replaced by canine serum (*Mussmann et al., 2003*) are informative. The BES1 TfR does not bind canine transferrin and cell proliferation arrested. After ~5 days, proliferating cells appeared in the culture having switched BES and thus TfR to one able to bind canine transferrin (*van Luenen et al., 2005*). In addition, supplementing the canine serum-based medium with bovine transferrin prevented arrest of proliferation indicating that it was due to lack of transferrin alone (*Mussmann et al., 2004*). In contrast to experiments that wash cells into solutions that do not support growth, the transfer of BES1-VSG221 cells to medium with canine serum starves the cells of just transferrin. On transfer from FBS to canine serum, the TfR abundance increased four- to fivefold and the receptor was present all over the cell surface (*Mussmann et al., 2003*). It is unlikely that the distribution of TfR on the cell surface results from a retention system being overwhelmed given the variation in TfR expression observed from different BESs (*Figure 1*) but is due to an absence of ligand-mediated uptake. Different TfRs have a wide range of affinities for transferrins ($K_D$ from as low as 1.4 nM to >5 μM; *Trevor et al., 2019*), and there will be an equilibrium with the relative amounts of TfR on the cell body surface compared with the flagellar pocket and internal structures that will vary with the identity of the TfR and the identity and concentration of transferrin.

The experiments here were informed by the kinetics of rapid transferrin uptake *Figure 3* and as an unpublished observation in *Mussmann et al., 2004*, we did not manipulate the cells before ligand uptake experiments or prior to fixation for immunolocalisation. The result was clear evidence that TfR is expressed over the whole cell surface and is not restricted to the flagellar pocket. This was reinforced by the localisation of the HpHbR and FHR, both of which are monomeric (*Lane-Serff et al., 2014*; *Macleod et al., 2020*), contain a single GPI-anchor, and are also present over the whole cell surface. We also observed variation in TfR and HpHbR abundance between individual cells, as has been seen previously for TfR (*Duncan et al., 2024*; *Steverding et al., 1994*), and this suggests sensitivity to the requirements at the individual cell level. Whether this relates to cell cycle stage or any developmental progression remains to be investigated. The expression of the TfR is sensitive to the availability of $Fe^{3+}$ with several fold increases in mRNA and protein induced by iron chelators (*Gilabert Carbajo et al., 2021*). Growth in medium dependent on rabbit serum resulted in high levels of expression of the FHR, this could result from a factor in rabbit serum inducing expression or failure to bind ligand.

The findings here still leave the question of how the trypanosome avoids receptor antibody-based clearance by the host immune system? Vaccination of mice with ISGs did not result in any protection (*Ziegelbauer and Overath, 1993*), this shows that receptors do not have to be sequestered in the flagellar pocket to avoid the host adaptive immune system. The effective killing of trypanosomes by antibody-drug conjugates recognising the HpHbR shows that receptors are accessible (*MacGregor et al., 2019*), as does inhibition of transferrin binding by Fab fragments (*Steverding et al., 1995*). It is likely that a combination of processes results in resistance to receptor antibody-mediated killing: first antibodies have to recognise parts of the receptor that are accessible in the VSG coat, this will be affected by the structure of the receptor and how much is occluded by ligand binding, a function of ligand concentrations, host species and receptor $K_D$. Enough antibody has to be present on the cell surface for long enough to elicit a response. Two factors affect this, first the copy number of the receptor, ISGs are present at 50,000–70,000 per cell (*Ziegelbauer et al., 1992*; <1% of VSG), TfR at a few thousand, but variable (*Steverding et al., 1995*), and HpHbR at 300–400 (*Vanhollebeke et al., 2008*). Second, the half time of the receptor antibody complex on the cell body plasma membrane will be short (*Steverding et al., 1995*) as the hydrodynamic forces rapidly push VSG-antibody complexes toward the flagellar pocket and endocytosis (*Engstler et al., 2007*). In summary, we provide evidence that GPI-anchored receptors in trypanosomes are not sequestered in the flagellar pocket but are distributed across the cell body and flagellum surface plasma membrane and this leads to re-evaluation of how trypanosomes avoid receptor-antibody-mediated clearance.

## Materials and methods

### Cell lines

*T. brucei* Lister427 bloodstream form cell lines were used throughout expressing either VSG221 (MiTat1.2) from BES1, VSG118 (MITat1.5) from BES3, or VSG224 (MITat1.3) from BES7. BES7 expressing cells were as described in *Hertz-Fowler et al., 2008* where BES1 and BES7 were tagged

with puromycin and G418 resistance genes, respectively. Cells were maintained in culture in HMI-11 medium (*Hirumi and Hirumi, 1989*). For ligand-uptake assays and immunofluorescence cells were harvested in mid-log phase ($0.5–1\times10^6$ cells/ml). To allow further modification by (CRISPR)-associated gene 9 (Cas9) gene editing, BES1 and BES7 expressing cell lines were first modified by transfection with a construct containing Cas9 and T7 RNA polymerase transgenes (*Beneke et al., 2017*). Antibiotic selection was used when appropriate. The primers used to generate the cell lines are detailed in *Supplementary file 2*.

## Constructs for ESAG7 +/- GPI attachment site

Constructs are represented schematically in *Figure 2A* and described in brief below:

### BES7 constructs

The presence of a G418 resistance gene upstream of ESAG7 was utilised to generate the required constructs from BES7 genomic DNA (*Hertz-Fowler et al., 2008*). A PacI-EcoRI PCR fragment from the promoter to the G418 resistance gene start was annealed to an EcoRI-PacI PCR fragment from the G418 stop codon through to the ESAG7 3′ UTR. The resulting EcoRI site was used to insert a blasticidin resistance open reading frame. The entire construct was cloned into pGEM-T, allowing the required cassette to be excised by PacI for transfection into BES7 cells expressing Cas9, alongside suitable sgRNAs to target the BES7-ESAG7 region, designed by ChopChop (https://chopchop.cbu.uib.no/). This construct contained the endogenous BES7 ESAG7 including its GPI anchor. To remove the GPI anchor addition sequences and express an ESAG7 similar to those in the other Lister427 bloodstream expression sites, a SacI-BsrBI fragment was excised and replaced with the corresponding fragment from BES1 ESAG7. This fragment was generated by RT-PCR from RNA isolated from BES1 VSG221 cells using standard methods.

### BES1 constructs

A similar approach was used to make ESAG7 modifying constructs from BES1 exploiting a puromycin resistance gene located between the promoter and the ESAG7 gene in the BES1 expression site (*Hertz-Fowler et al., 2008*). PacI-BstEII and BstEII-PacI PCR fragments were generated from genomic DNA, ligated and cloned into pGEM-T. To generate BES1-ESAG7 with a GPI anchor from the BES1 expression site, the distal SacI-PacI fragment was replaced by a PCR fragment encoding SacI-ESAG7 stop codon (blunt end) from BES7-ESAG7 DNA and a BES1-ESAG7 3′ UTR region from the BES1 site corresponding to the BES7-ESAG7 stop codon (blunt end) to PacI. Constructs were linearised with PacI and transfected with sgRNAs targeting the ESAG7 gene in either BES7 or BES1 and electroporated into BES1 VSG221 cells expressing Cas9. BES7 modified cell lines were selected with blasticidin and BES1 modified cell lines with puromycin.

### Changing the VSG from 224 to 221

BES7-VSG224 expression was switched to VSG221 by transfection of a linearised plasmid fragment encoding VSG221, flanked by 5′ and 3′ VSG221 intergenic regions and a downstream G418 resistance gene flanked by alpha-beta tubulin intergenic regions. Sequences targeting the 5′ and 3′ UTRs of the endogenous VSG224 gene were generated by PCR from BES7 (TAR153) DNA (a kind gift of Luísa Figueiredo) and cloned via KpnI/XhoI and SpeI/SacI fragments upstream and downstream of the VSG221-blasticidin expression cassette. G418-resistant clones were screened for binding to anti-VSG221 antibody through immunofluorescence assays and confirmed by western blot analysis.

## Protein gel electrophoresis and western blotting

Protein samples were prepared and analysed by SDS-PAGE and western blotting following standard methods. Briefly, cells were harvested by centrifugation, washed in serum-free HMI-9, and resuspended in Protein Sample Buffer (10% glycerol, 2% w/v SDS, 80 mM Tris-HCl pH 6.8), at a concentration of $2\times10^8$ cells/ml. Immobilon-P membrane (Millipore) was used to retain the proteins. The primary antibodies used during western blotting are rabbit anti-FHR (*Macleod et al., 2020*) and rabbit anti-TfR (a kind gift of Piet Borst). Polyclonal anti-VSG221 and anti-SCD6 antibodies were generated from recombinant *E. coli* expressed proteins, injected into rabbits by standard methods (Covalab). Secondary antibodies used were Peroxidase-conjugated AffiniPure Goat Anti-Rabbit IgG (H+L;

Jackson Immunoresearch) and Alexa Fluor 488-Donkey Anti-Rabbit IgG (H+L; Invitrogen). Membranes were blocked in Tris-buffered saline (TBS), 0.1% Tween20, 2% w/v Marvel dried milk and probed in appropriate dilutions of primary and secondary antibodies. Washing steps were performed with TBS, 0.1% Tween20 and blots were analysed on a Bio-Rad ChemiDoc system, ensuring signals remained below saturation limits. Quantitation of transferrin receptor protein level was performed using Fiji-ImageJ software (https://imagej.net/software/fiji/), with background subtraction and comparison with intensity measurements taken from a protein dilution series from BES7-expressing cells.

## RNAseq

mRNA was sequenced using Illumina paired-end sequencing technology by the Beijing Genomics Institute. Raw reads were deposited to ENA under the accession number PRJEB90063. Analysis was performed using Galaxy (https://www.usegalaxy.org; *Afgan et al., 2018*). Reads were quality filtered using Trimmomatic (*Bolger et al., 2014*). Sequences were searched for all common Illumina adaptors and the settings used were: LEADING:10 TRAILING:10 SLIDINGWINDOW:5:15 MINLEN:50. mRNA abundance was calculated using SalmonQuant with default settings and *Trypanosoma brucei* Lister strain 427 2018 v68 reference transcriptome and genomes from TriTrypDB (*Aslett et al., 2010*).

## Ligand uptake

Bovine holo-transferrin (Thermo Fisher) was dialyzed against PBS and labelled with Alexa fluor 568 NHS ester (Thermo Fisher) following the manufacturer's protocol. Canine transferrin was obtained from Abbexa and loaded with iron using a fourfold molar excess of ammonium iron (III) sulphate in presence of 5 mM sodium bicarbonate followed by dialysis against 10 mM HEPES pH 7.4, 150 mM NaCl as described previously (*Trevor et al., 2019*). Canine holo-transferrin was then labelled with Alexa fluor 568 NHS ester (Thermo Fisher) as above. The Hp-Hb complex was purified as described (*Lane-Serff et al., 2014*) and labelled with Alexa fluor 568 NHS ester (Thermo Fisher) following the manufacturer's protocol.

Mid log-phase cells (~6 x $10^5$ /ml) were supplemented with 500 nM Alexa 568 labelled ligand and incubated at 37 °C. At time points, cells were fixed by the addition of 8% formaldehyde in PBS to a final concentration 1% and incubated for 5 min. Cells were further washed with PBS plus 1% BSA and DNA was stained with Hoechst 33258. Slides were prepared and images were captured within the next 30 min using Zeiss Axio Imager.Z2 with ×100 objective. Samples in any one experiment were excited with the same intensity and using the same exposure time. Images were processed using the ZEN Blue software.

## Immunofluorescence localisation and microscopy

*T. brucei* cells were fixed in culture by the addition of one volume of 8% formaldehyde, 0.4% glutar-aldehyde (EM grade TAAB Laboratories Equipment Ltd) in PBS for 10 min. After fixation, cells were washed with serum-free HMI-9 plus 1% bovine serum albumin (BSA) twice and immobilised on poly-lysine-coated slides for 1 hr. Immobilised cells were treated with 0.1% Triton X-100 (in PBS) for 5 min, washed with PBS and then incubated with PBS, 1% BSA for 10 min. Cells were incubated with the rabbit anti-TfR (1:50) or anti-FHR (*Macleod et al., 2020*; 1:50) in PBS, 1% BSA for 1 hr, washed three times with PBS and then with donkey anti-rabbit Alexa488 (1:800) in PBS, 1% BSA for 1 hr. Cells were washed three times with PBS and the last wash contained 0.1 µg/ml Hoechst 33258 to stain DNA. Cells were mounted with anti-fade and were visualised under Zeiss Axio Imager Z2 with X100 objective. In any one experiment, samples were illuminated with the same exposure time, intensity for respective excitation channels. All immunofluorescence experiments were repeated three to six times with similar results. The images were processed with ZEN Blue software by adjusting the brightness and contrast with the same changes being made to all images in any one experiment including the negative controls. Individual cells were manually outlined to define the whole-cell regions of interest (ROI), and background fluorescence was measured from cell-free regions within the same field using the ZEN Blue software and it was subtracted. For each condition, mean fluorescence intensity (arbitrary unit) was quantified from at least 50–100 cells across the repeated experiments. Mean fluorescence intensity was measured and background fluorescence intensity was subtracted. ImageJ software was used to measure the point fluorescence intensity of ligands and receptor antibodies by drawing a line perpendicular to the long axis of the cell, and the distribution profile against the

distance was plotted. For 3D structured illumination, a Zeiss Elyra7 microscope with X63 objective was used in the Cambridge Advanced Imaging Centre. Slides were prepared as above and the processing of all images and the 3D reconstruction were performed with ZEN Blue software only by adjusting the brightness contrast. All raw and processed images are available at EMBL-EBI BioImage Archive accession S-BIAD2480.

### Cell line availability

All cell lines are available on request.

## Acknowledgements

We thank: Luísa Figueiredo for the gift of the cell line expressing VSG224 from BES7 and a BES7 TAR clone; Piet Borst for the transferrin receptor antibodies; Jay Bangs for a lot of fun discussions (but no fish); and the Cambridge Advance Imaging Centre for Super Resolution Microscopy. MC and MH are Wellcome Investigators (MC, 217138/Z/19/Z; MH, 220797/Z/20/Z). Initial funding for SB was from the Newton-Bhabha Fund.

## Additional information

### Funding

| Funder | Grant reference number | Author |
|---|---|---|
| Newton Fund | Newton-Bhabha Fund | Sourav Banerjee |
| Wellcome Trust | 10.35802/217138 | Mark Carrington |
| Wellcome Trust | 10.35802/220797 | Matthew K Higgins |

The funders had no role in study design, data collection and interpretation, or the decision to submit the work for publication. For the purpose of Open Access, the authors have applied a CC BY public copyright license to any Author Accepted Manuscript version arising from this submission.

### Author contributions

Sourav Banerjee, Data curation, Software, Formal analysis, Validation, Investigation, Visualization, Methodology, Writing – original draft, Writing – review and editing; Nicola Minshall, Data curation, Software, Formal analysis, Validation, Investigation, Methodology; Alexander D Cook, Olivia Macleod, Helena Webb, Data curation, Validation, Investigation; Matthew K Higgins, Supervision, Funding acquisition, Writing – review and editing; Mark Carrington, Conceptualization, Formal analysis, Supervision, Funding acquisition, Methodology, Writing – original draft, Project administration, Writing – review and editing

### Author ORCIDs

Sourav Banerjee ⓘ https://orcid.org/0000-0002-0716-3668
Nicola Minshall ⓘ https://orcid.org/0000-0002-8350-4903
Alexander D Cook ⓘ https://orcid.org/0000-0003-1283-1541
Olivia Macleod ⓘ https://orcid.org/0000-0002-5747-8019
Matthew K Higgins ⓘ https://orcid.org/0000-0002-2870-1955
Mark Carrington ⓘ https://orcid.org/0000-0002-6435-7266

Reviewer #1 (Public review): https://doi.org/10.7554/eLife.107191.3.sa1
Reviewer #2 (Public review): https://doi.org/10.7554/eLife.107191.3.sa2
Author response https://doi.org/10.7554/eLife.107191.3.sa3

## Additional files

### Supplementary files

Supplementary file 1. G and ESAG6/7 mRNA abundances determined by analysis of RNAseq data.

Supplementary file 2. Oligonucleotide sequences.

Supplementary file 3. Data for fluorescence measurements.

Supplementary file 4. Data for fluorescence measurements.

Supplementary file 5. Data for fluorescence measurements.

MDAR checklist

### Data availability

All data generated or analyzed during this study are included in the manuscript and supporting files; source data files have been provided for all figures. RNA sequence data were deposited to ENA under the accession number PRJEB90063. All raw and processed microscopic images are available at EMBL-EBI BioImage Archive accession S-BIAD2480.

The following datasets were generated:

| Author(s) | Year | Dataset title | Dataset URL | Database and Identifier |
|---|---|---|---|---|
| Banerjee S, Minshall N, Cook AD, Macleod O, Webb H, Carrington MHM | 2026 | RNAseq reads from *Trypanosoma brucei* Lister427 strains expressing VSG221 from BES1, VSG118 from BES3 and VSG224 from BES7 | https://www.ebi.ac.uk/ena/browser/view/PRJEB90063 | EBI European Nucleotide Archive, PRJEB90063 |
| Banerjee S, Minshall N, Cook AD, Macleod O, Webb H, Higgins M, Carrington M | 2026 | Cell surface localisation of GPI-anchored receptors in *Trypanosoma brucei* | https://www.ebi.ac.uk/biostudies/BioImages/studies/S-BIAD2480?query=S-BIAD2480 | EMBL-EBI BioImage Archive, S-BIAD2480 |

The following previously published dataset was used:

| Author(s) | Year | Dataset title | Dataset URL | Database and Identifier |
|---|---|---|---|---|
| Hertz-Fowler C, Figueiredo LM, Quail MA, Becker M, Jackson A, Bason N, Brooks K, Churcher C, Fahkro S, Goodhead I, Heath P, Kartvelishvili M, Mungall K, Harris D, Hauser H, Sanders M, Saunders D, Seeger K, Sharp S, Berriman M | 2008 | *Trypanosoma brucei* Lister 427 surface glycoprotein expression site BES1/TAR40, from bloodstream | https://www.ebi.ac.uk/ena/browser/view/FM162566 | EMBL-EBI BioImage Archive, FM162566 |

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
