## [Editor Report · eLife Assessment]

This **valuable** manuscript investigates the localisation of nutrient receptors in bloodstream stage trypanosomes, with implications for both nutrient uptake and immune evasion. Results after direct fixation of the cells in culture medium (as opposed to fixation after centrifugation) provide **compelling** evidence that the amounts of receptors on the surface of the cell, as opposed to the flagellar pocket, have previously been severely underestimated.

---

## [Referee Report · Reviewer #1 (Public review)]

Summary:

An interesting manuscript from the Carrington lab is presented investigating the behavior of single vs double GPI-anchored nutrient receptors in bloodstream form (BSF) *T. brucei*. These include the transferrin receptor (TfR), the HpHb receptor (HpHbR), and the factor H receptor (FHR). The central question is why these critical proteins are not targeted by host acquired immunity. It has generally been thought that they are sequestered in the flagellar pocket (FP), where they are subject to rapid endocytosis - any Ab:receptor complexes would be rapidly removed from the cell surface. This manuscript challenges that assumption by showing that these receptors can be found all over the outer cell body and flagella surfaces - if one looks in an appropriate manner (rapid direct fixation in culture media).

Strengths and weaknesses:

(1) The presence of a second ESAG6 gene in the BES7 expression site was noted in the previous review. This is now noted and discussed appropriately in the current version.

(2) Surface binding studies: The ability of cells to bind tagged-Tf while in complete media was challenged and it was suggested that classic competition studies be performed to validate saturable ligand binding. This has been done now and the results confirm that this is so. A reasonable discussion of the results is presented.

(3) Variable TfR expression in different BESs: The claim that specific ES environment is the dominant factor controlling TfR expression levels was challenged in that the presented results could be due to technical issues. RNA seq has now been performed confirming that the differences in TfR abundance is indeed directly related to mRNA levels

(4) Surface immuno-localization of receptors: In regard to the novel immunofluorescence (direct fixation) methodology used to demonstrate TfR on the cell surface the authors were asked of they had attempted more traditional methods that involve centrifugation/washing. These data are now provided (Fig S5) and do indicate that centrifugation does reduce signal, likely due to shedding and/or internalization during the procedure. Nevertheless, significant signal is present after centrifugation leaving the issue of why others have never detected significant surface TfR.

These responses address all the major concerns with the original submission and a greatly improved manuscript is now submitted.

---

## [Referee Report · Reviewer #2 (Public review)]

The revised data support the conclusion that methodological differences can influence apparent receptor localization. However, key claims regarding functional surface engagement of TfR and hydrodynamic clearance remain based largely on indirect evidence and model-based interpretation. These conclusions should therefore be phrased more cautiously.

I thank the authors for their careful rebuttal and the additional experiments included in the revised manuscript. The new fixation comparisons and transferrin competition assays substantially strengthen the technical basis of the study and address several of the original concerns.

However, some conclusions remain more inferential than directly supported by the data. While the fixation and washing controls demonstrate that methodology influences apparent TfR localisation, they do not directly establish that previous protocols quantitatively redistribute surface TfR into the flagellar pocket. Statements implying such redistribution should therefore be phrased more cautiously.

Similarly, the added transferrin binding controls argue against non-specific interactions, but functional engagement of surface-exposed TfR in intact bloodstream-form parasites remains supported mainly by indirect evidence. The proposed explanation involving rapid on/off rates and newly arriving receptors is plausible but should be more clearly identified as an inference.

---

## [Author Response]

The following is the authors’ response to the original reviews.

**Public Reviews:**

**Reviewer #1 (Public review):**
Summary:An interesting manuscript from the Carrington lab is presented investigating the behavior of single vs double GPI-anchored nutrient receptors in bloodstream form (BSF) *T. brucei*. These include the transferrin receptor (TfR), the HpHb receptor (HpHbR), and the factor H receptor (FHR). The central question is why these critical proteins are not targeted by host-acquired immunity. It has generally been thought that they are sequestered in the flagellar pocket (FP), where they are subject to rapid endocytosis - any Ab:receptor complexes would be rapidly removed from the cell surface. This manuscript challenges that assumption by showing that these receptors can be found all over the outer cell body and flagella surfaces, if one looks in an appropriate manner (rapid direct fixation in culture media).The main part of the manuscript focuses on TfR, typically a GPI1 heterodimer of very similar E6 (GPI anchored) and E7 (truncated, no GPI) subunits. These are expressed coordinately from 15 telomeric expression sites (BES), of which only one can be transcribed at a time. The authors identify a native E6:E7 pair in BES7 in which E7 is not truncated and therefore forms a GPI2 heterodimer. By in situ genetic manipulation, they generate two different sets of GPI1:GPI2 TfR combinations expressed from two different BESs (BES1 and BES7). Comparative analyses of these receptors form the bulk of the data.The main findings are:(1) Both GPI1 and GPI2 TfR can be found on the cell body/flagellar surface.(2) Both are functional for Tf binding and uptake.(3) GPI2 TfR is expressed at ~1.5x relative to GPI1 TfR(4) Ultimate TfR expression level (protein) is dependent on the BES from which it is expressed.Most of these results are quite reasonably explained in light of the hydrodynamic flow model of the Engstler lab and the GPI valence model of the Bangs lab. Additional experiments, again by rapid fixation, with HpHbR and FHR, show that these GPI1 receptors can also be seen on the cell surface, in contrast to published localizations.It is quite interesting that the authors have identified a native GPI2 TfR. However, essentially all of the data with GPI2 TfR are confirmatory for the prior, more detailed studies of Tiengwe et al. (2017). That said, the suggestion that GPI2 was the ancestral state makes good evolutionary sense, and begs the question of why trypanosomes prefer GPI1 TfR in 14 of 15 ESs (i.e., what is the selection pressure?)Strengths and weaknesses:(1) BES7 TfR subunit genes (BES7_Tb427v10): There are actually three (in order 5'3'): E7gpi, E6.1 and E6.2. E6.1 and E6.2 have a single nucleotide difference. This raises the issue of coordinate expression. If overall levels of E6 (2 genes) are not down-regulated to match E7 (1 gene), this will result in a 2x excess of E6 subunits. The most likely fate of these is the formation of non-functional GPI2 homodimers on the cell surface, as shown in Tiengwe et al. (2017), which will contribute to the elevated TfR expression seen in BES7.

We would like to thank the reviewer for pointing out that there are two ESAG6 genes in BES7, we had relied on the publicly available annotation and should have known better.

For transferrin expression levels, see the discussion in response to reviewer 1 point 3 below

(2) Surface binding studies: This is the most puzzling aspect of the entire manuscript. That surface GPI2 TfR should be functional for Tf binding and uptake is not surprising, as this has already been shown by Tiengwe et al. (2017), but the methodology for this assay raises important questions. First, labeled Tf is added at 500 nM to live cells in complete media containing 2.5 uM unlabeled Tf - a 5x excess. It is difficult to see how significant binding of labeled TfR could occur in as little as 15 seconds under these conditions.

The k_on_ for transferrin is very rapid BES1 TfR / bovine transferrin at pH7.4 = 4.5 x 10^5^ M^-1^s^-1^ (Trevor et al., 2019) and binding would occur to unoccupied receptors within 15 sec. The k_off_ is also fast BES1 TfR / bovine transferrin at pH7.4 = 3.6 x 10^-2^ s^-1^ (Trevor et al., 2019) and there would be exchange of transferrin within the time taken for endocytosis. These values are *in vitro* with purified proteins, the *in vivo* values may be affected by the VSG coat.

The failure to bind canine transferrin (Supp. Figure 4B) acts as a control for specificity of the interaction.

We have now performed a competition experiment as an additional control; cells in culture were supplemented with: A, 0.5 µM labelled transferrin; B, 0.5 µM labelled and 2.5 µM unlabelled transferrin; C, 0.5 µM labelled and 5 µM unlabelled transferrin, fixed after 60 s and visualised by fluorescence microscopy (Figure S4C). There was effective competition and greatly reduced binding of transferrin was seen in the presence of a 10-fold excess of unlabelled. We would like to thank the reviewer for suggesting this experiment.

Second, Tiengwe et al. (2017) found that trypanosomes taken directly from culture could not bind labeled Tf in direct surface labelling experiments. To achieve binding, it was necessary to first culture cells in serum-free media for a sufficient time to allow new unligated TfR to be synthesized and transported to the surface. This result suggests that essentially all surface TfR is normally ligated and unavailable to the added probe.

As part of the preliminary experiments for this paper we found that centrifugation followed by resuspension in either complete or serum free (but 1% BSA) medium resulted in a reduction is total cellular TfR and determined by western blotting. We have now included this experiment (Figure S4D). The inference from this experiment is that centrifugation and subsequently incubation will have an effect on receptor detection and endocytosis rates for a discreet time period.

The amount of binding of labelled transferrin to cells in culture will depend on the specific activity of the labelled transferrin. This reasoning was behind the use of 0.5 µM labelled transferrin when roughly 1 in 6 molecules in the culture medium are labelled and there was only a small effect on the overall concentration of transferrin.

Third, the authors have themselves argued previously, based on binding affinities, that all surface-exposed TfR is likely ligated in a natural setting (DOI:10.1002/bies.202400053). Could the observed binding actually be non-specific due to the high levels of fixative used?

The absence of binding/uptake of canine transferrin argues against a non-specific interaction. In our previous publication, we did not pay enough attention to the on and off rates which allow for a degree of exchange and, here, TfR newly appearing on the cell surface has a 1 in 6 chance of binding a labelled transferrin.

(3) Variable TfR expression in different BESs: It appears that native TfR is expressed at higher levels from BES7 compared to BES1, and even more so when compared to BES3. This raises the possibility that the anti-TfR used in these experiments has differential reactivity with the three sets of TfRs. The authors discount this possibility due to the overall high sequence similarities of E6s and E7s from the various ESs. However, their own analyses show that the BES1, BES3, and BES7 TfRs are relatively distal to each other in the phylogenetic trees, and this Reviewer strongly suspects that the apparent difference in expression is due to differential reactivity with the anti-TfR used in this work. In the grand scheme, this is a minor issue that does not impact the other major conclusions concerning TfR localization and function, nor the behavior of HpHbR and FHR. However, the authors make very strong conclusions about the role of BESs in TfR expression levels, even claiming that it is the 'dominant determinant' (line 189).

This point is valid but exceptionally difficult to address at the protein level. As an orthogonal approach, we performed RNAseq analysis of the ‘wild type’ BES1, BES3, and BES7 cell lines to determine whether differences in receptor mRNA levels were consistent with the proposed difference in protein levels (Table S1). The analysis showed total ESAG6/7 mRNA levels to vary in a similar manner to the protein estimates with BES3 < BES1 < BES7 providing support for the differences in protein levels.

The strongest evidence for the expression site determining the TfR level is the comparison of the cell lines in which the VSG were exchanged. This had no effect on TfR levels and so there is no evidence that the identity of the VSG alters TfR expression.

(4) Surface immuno-localization of receptors: These experiments are compelling and useful to the field. To explain the difference with essentially all prior studies, the authors suggest that typical fixation procedures allow for clearance of receptor:ligand complexes by hydrodynamic flow due to extended manipulation prior to fixation (washing steps). Despite the fact that these protocols typically involve ice-cold physiological buffers that minimize membrane mobility, this is a reasonable possibility. Have the authors challenged their hypothesis by testing more typical protocols themselves? Other contributing factors that could play a role are the use of deconvolution, which tends to minimize weak signals, and also the fact that investigators tend to discount weak surface signals as background relative to stronger internal signals.

We have added preliminary experiments that compared fixation protocols in two parts. First the effect on TfR levels of washing and resuspending cells discussed above (Figure S4D), and second how different fixation protocols alter apparent TfR immunolocalisation (Supp Figure S5A-B). The comparison shows that both the absence of glutaraldeyde and the use of washing alters the outcome.

(5) Shedding: A central aspect of the GPI valence model (Schwartz et al., 2005, Tiengwe et al., 2017) is that GPI1 reporters that reach the cell body surface are shed into the media because a single dimyristoylglycerol-containing GPI anchor does not stably associate with biological membranes. As the authors point out, this is a major factor contributing to higher steady-state levels of cell-associated GPI2 TfR relative to GPI1 TfR. Those studies also found that the size/complexity of the attached protein correlated inversely with shedding, suggesting exit from the flagellar pocket as a restricting factor in cell body surface localization. The amount of newly synthesized TfR shed into the media was ~5%, indicating that very little actually exits the FP to the outer surface. In this regard, is it possible to know the overall ratio of cell surface:FP:endosomal localized receptors? Could these data not be 'harvested' from the 3D structural illumination imaging?

A ratio could be determined but we did not do this as it would only be valid if the antibody has equal access to the internal TfR in a diluted VSG environment and the external VSG embedded in a densely packed and cross-linked VSG layer As such, we would have no confidence in the accuracy of any estimate.

**Reviewer #2 (Public review):**
The work has significant implications for understanding immune evasion and nutrient uptake mechanisms in trypanosomes.While the experimental rigor is commendable, revisions are needed to clarify methodological limitations and to broaden the discussion of functional consequences.The authors argue that prior studies missed surface-localized TfR due to harsh washing/fixation (e.g., methanol). While this is plausible, additional evidence would strengthen the claim.

Preliminary experiments that compared fixation protocols are now included to show that method affects outcome.

It remains unclear how centrifugation steps of various lengths (as in previous publications) can equally and quantitatively redistribute TfR into the flagellar pocket. If this were the case, it should be straightforward for the authors to test this experimentally.

Not aware of previous studies that demonstrate equal and quantitative redistribution to the flagellar pocket. In previous reports, there is variation in cell surface/flagellar pocket localisation depending on expression levels, for example (Mussmann et al., 2003) (Mussmann et al., 2004), it’s worth noting that the increase in TfR expression in these papers is similar to the difference in the cell lines used here. In addition, most report the presence of TfR in endosomal compartments. In the experiments here, there are cells where the majority of signal from labelled transferrin is present in the flagellar pocket and the argument is that this is a stage of a continuous process in which the receptor picks up a transferrin on the cell surface and is swept towards the pocket.

If TfR is distributed over the cell surface, live-cell imaging with fluorescent transferrin should be performed as a control. Modern detection limits now reach the singlemolecule level, and transient immobilization of live trypanosomes has been established, which would exclude hydrodynamic surface clearance as a confounding factor.

This is non-trivial and is a longer-term aim. The immobilisation involves significant manipulation of the cells prior to restraining.

In most images, TfR is not evenly distributed on the surface but rather appears punctate. Could this reflect localization to membrane domains? Immuno-EM with high-pressure frozen parasites could resolve this question and is relatively straightforward.

There is a non-uniform appearance in the super-resolution images for both TfR and FHR. We cannot distinguish whether this represents random variation in receptor density over the cell surface or results from a biological phenomenon. Whatever the cause, the experiments showed unambiguous cell surface localisation.

The authors might consider discussing whether differences in parasite life cycle stages (procyclic versus bloodstream forms) or culture conditions (e.g., cell density) affect localization. The developmentally regulated retention of GPI-anchored procyclin in the flagellar pocket might be worth mentioning.

The aim of this paper was to determine the localisation of receptors in proliferating bloodstream form trypanosomes in culture. TfR and HpHbR are not expressed in insect stages in culture. FHR is expressed in insect stages and is present all over the cell surface (Macleod et al., 2020). A procyclin-based reporter was distributed over the whole cell surface in one report (Schwartz et al. 2005). In other reports, the retention of procyclin in the flagellar pocket of proliferating bloodstream forms is probably dependent on structure/sequence as other single GPI-anchored proteins, such as FHR (Macleod et al., 2020) and GPI-anchored sfGFP (Martos-Esteban et al., 2022) can access the surface.

References:

MacGregor, P., Gonzalez-Munoz, A. L., Jobe, F., Taylor, M. C., Rust, S., Sandercock, A. M., Macleod, O. J. S., Van Bocxlaer, K., Francisco, A. F., D’Hooge, F., Tiberghien, A., Barry, C. S., Howard, P., Higgins, M. K., Vaughan, T. J., Minter, R., & Carrington, M. (2019). A single dose of antibody-drug conjugate cures a stage 1 model of African trypanosomiasis. PLoS Neglected Tropical Diseases, 13(5), e0007373. https://doi.org/10.1371/journal.pntd.0007373

Macleod, O. J. S., Bart, J.-M., MacGregor, P., Peacock, L., Savill, N. J., Hester, S., Ravel, S., Sunter, J. D., Trevor, C., Rust, S., Vaughan, T. J., Minter, R., Mohammed, S., Gibson, W., Taylor, M. C., Higgins, M. K., & Carrington, M. (2020). A receptor for the complement regulator factor H increases transmission of trypanosomes to tsetse flies. Nature Communications, 11(1), 1326. https://doi.org/10.1038/s41467-020-15125-y

Martos-Esteban, A., Macleod, O. J. S., Maudlin, I., Kalogeropoulos, K., Jürgensen, J. A., Carrington, M., & Laustsen, A. H. (2022). Black-necked spitting cobra (Naja nigricollis) phospholipases A2 may cause *Trypanosoma brucei* death by blocking endocytosis through the flagellar pocket. Scientific Reports, 12(1), 6394. https://doi.org/10.1038/s41598-02210091-5

Mussmann, R., Engstler, M., Gerrits, H., Kieft, R., Toaldo, C. B., Onderwater, J., Koerten, H., van Luenen, H. G. A. M., & Borst, P. (2004). Factors affecting the level and localization of the transferrin receptor in *Trypanosoma brucei*. The Journal of Biological Chemistry, 279(39), 40690–40698. https://doi.org/10.1074/jbc.M404697200

Mussmann, R., Janssen, H., Calafat, J., Engstler, M., Ansorge, I., Clayton, C., & Borst, P. (2003). The expression level determines the surface distribution of the transferrin receptor in *Trypanosoma brucei*. Molecular Microbiology, 47(1), 23–35. https://doi.org/10.1046/j.13652958.2003.03245.x

Schwartz, K. J., Peck, R. F., Tazeh, N. N., & Bangs, J. D. (2005). GPI valence and the fate of secretory membrane proteins in African trypanosomes. Journal of Cell Science, 118(Pt 23), 5499–5511. https://doi.org/10.1242/jcs.02667

Trevor, C. E., Gonzalez-Munoz, A. L., Macleod, O. J. S., Woodcock, P. G., Rust, S., Vaughan, T. J., Garman, E. F., Minter, R., Carrington, M., & Higgins, M. K. (2019). Structure of the trypanosome transferrin receptor reveals mechanisms of ligand recognition and immune evasion. Nature Microbiology, 4(12), 2074–2081. https://doi.org/10.1038/s41564-019-0589-0

**Recommendations for the authors:**

**Reviewer #1 (Recommendations for the authors):**
Major Recommendations:(1) 2 E6 gene in BES7s: This does not affect the overall conclusions, but the text should be modified to reflect the existence of the second gene, and to discuss the ramifications.

This has been corrected

(2) Surface binding studies: To clarify this issue, two experimental approaches are strongly recommended. First: additional excess unlabelled Tf should be added. If binding is truly receptor-mediated, it must by definition be saturable at some experimentally achievable level. Second: TfR expression should be abrogated by RNAi silencing to show that binding is TfR-dependent. Without some validation of specific binding by one or both of these approaches, these counter-intuitive results must be questioned.

The excess unlabelled transferrin experiment is now included (we would like to thank the reviewer for this suggestion). The absence of binding of canine transferrin provides strong evidence for the specificity.

(3) Variable TfR expression in different BESs: To make such claims, quantitative RTPCR should be performed with conserved primers to assess the actual relative expression at the transcriptional level. Absent this, the claims should be eliminated, or at the very least greatly tempered.

This has been done using an RNAseq analysis.

(4) Surface immuno-localization of receptors: An example of discounting weak signals as background can be seen in Figure 8 of Duncan et al. (2024). It has also been shown that at least one other GPI1 reporter (procyclin) is readily detected on the outer cell surface under ectopic expression in BSF trypanosomes (Schwartz et al., 2005) using typical fixation procedures. This could be cited, and the authors could discuss the fact that procyclin is not a receptor and may not be susceptible to hydrodynamic drag.

Yes

Minor issues:(1) Fully appreciating the data presented requires an understanding of the hydrodynamic flow and GPI valence models of the Engstler and Bangs labs, respectively. For the uninitiated,d it might perhaps be useful to include brief summaries of each in the Introduction.

Added to the introduction

(2) Lines 110-112: ISG65 and ISG75 both have strong localizations in endosomal compartments. This should be noted with citation of any of the work from the Field lab.

Added

(3) Lines 121-132: This passage presents the role of GPI anchors (1 vs 2) in a rather digital manner (in or out). Schwartz et al (2005) present a much more nuanced view of what is likely taking place. This is one reason summaries of hydrodynamic flow and GPI valence would be helpful.

Modified

(4) Lines 182-184: The increased size of GPI-anchored E7 is in part due to the presence of the GPI itself, as the authors state, but there are also 24 additional amino acid residues in this protein that contribute.

Modified

(5) Lines 212-214: Do p>0.95 and p>0.99 indicate statistical significance? This must be a typo.

Thank you, corrected

(6) Lines 218-219: The better references documenting GPI number in regard to turnover/shedding are Schwartz et al. 2005 and Tiengwe et al. 2017.

Changed

(7) Line 241 and Figures 3, 4, and 6: The transverse sections add little to the presentation. That there is signal variation in all dimensions is readily apparent from the images themselves, and similar profiles would be obtained regardless of the transect. Was there some process/rationale in the selection of the individual transects intended to make a broader point? If so, a description of the process should be provided.

The point was to show that the signal had a pattern consistent with plasma membrane (two distal peaks) as opposed to cytoplasm (single central peak). As such, we think it is important.

(8) Lines 582-596: Methodology for quantitation of cellular fluorescent signals should be provided.

Has been expanded

**Reviewer #2 (Recommendations for the authors):**
(1) As a less critical but still useful control, antibody accessibility assays on live versus fixed parasites could test whether VSG coats limit detection.

This could only be quantified by using a range of monoclonal antibodies which are not available.

(2) The rapid transferrin uptake (15-60 seconds) could reflect fast endocytic recycling rather than stable surface residency. A pulse-chase experiment tracking receptor movement would clarify this (though I acknowledge that this is technically challenging).

We agree that endocytic recycling is probably the main source of unoccupied TfR on the cell surface. It is hard to see how the pulse chase experiment could be performed without centrifugation which will affect the outcome – see above.

(3) Statistical and quantitative reporting

Added as Table S2- S4

(4) Report confidence intervals (e.g., for fluorescence intensity comparisons in Figure 3B) to contextualize claims of "no significant difference."

We do not claim ‘no significant difference’ and the SD overlap due to a high level of variation in the population

(5) Specify the number of biological replicates and cells analyzed per condition in the figure legends.

Added

(6) The study notes that surface-exposed receptors avoid antibody detection, but does not explore how.

We don’t claim that receptors avoid detection and have published evidence to the contrary. The cell has evolved mechanisms to reduce/minimise the effect of antibody binding.

(7) Comparing antibody binding to TfR in VSG221 versus VSG224 coats.

This is already present in Figure 3D

(8) Testing whether receptor shedding or conformational masking contributes to immune evasion.

A lifetime’s work

(9) Evolutionary trade-offs: Discuss why *T. brucei* maintains ~15 TfR variants if the GPI-anchor number has minimal impact on function (Figure 3).

The possible reason for the evolution of ~15 TfR variants was discussed in a previous publication.

(10) How do their findings align with recent studies on ISG75 surface exposure?

If this refers to the finding that ISG75 is an Ig Fc receptor, this has been included

(11) Add scale bars to 3D reconstructions (Figure 5).

Added

(12) Include a schematic summarizing key findings in the main text.

Chosen not to do

(13) Explicitly state where raw microscopy images, flow cytometry data, and analysis scripts are deposited.

Microscope Images have deposited in Bioimage Archive repository at EMBL/EBI No flow cytometry used

(14) Correct inconsistent GPI-anchor terminology (e.g., "glycosylphosphoinositol" to "glycosylphosphatidylinositol").

Our typo, corrected

(15) Clarify ambiguous phrases (e.g., "subtle mechanisms" in the Discussion).

Corrected